# Metabolic resistance of Aβ3pE-42, a target epitope of the anti-Alzheimer therapeutic antibody, donanemab

Nobuhisa Iwata[1,2], Satoshi Tsubuki[2], Misaki Sekiguchi[2], Kaori Watanabe-Iwata[1], Yukio Matsuba[2], Naoko Kamano[2], Ryo Fujioka[2], Risa Takamura[2], Naoto Watamura[2], Naomasa Kakiya[2], Naomi Mihira[2], Takahiro Morito[2], Keiro Shirotani[1], David MA Mann[3], Andrew C Robinson[3], Shoko Hashimoto[2], Hiroki Sasaguri[2], Takashi Saito[4,5], Makoto Higuchi[6], Takaomi C Saido[2]

**The amyloid β peptide (Aβ), starting with pyroglutamate (pE) at position 3 and ending at position 42 (Aβ3pE-42), predominantly accumulates in the brains of Alzheimer's disease. Consistently, donanemab, a therapeutic antibody raised against Aβ3pE-42, has been shown to be effective in recent clinical trials. Although the primary Aβ produced physiologically is Aβ1-40/42, an explanation for how and why this physiological Aβ is converted to the pathological form remains elusive. Here, we present experimental evidence that accounts for the aging-associated Aβ3pE-42 deposition: Aβ3pE-42 was metabolically more stable than other Aβx-42 variants; deficiency of neprilysin, the major Aβ-degrading enzyme, induced a relatively selective deposition of Aβ3pE-42 in both APP transgenic and App knock-in mouse brains; Aβ3pE-42 deposition always colocalized with Pittsburgh compound B–positive cored plaques in APP transgenic mouse brains; and under aberrant conditions, such as a significant reduction in neprilysin activity, aminopeptidases, dipeptidyl peptidases, and glutaminyl-peptide cyclotransferase-like were up-regulated in the progression of aging, and a proportion of Aβ1-42 may be processed to Aβ3pE-42. Our findings suggest that anti-Aβ therapies are more effective if given before Aβ3pE-42 deposition.**

## Introduction

Senile plaques, which are one of the key pathological features of Alzheimer's disease (AD), are composed of amyloid β peptides (Aβs) with a variety of secondary and tertiary structures because of post-translational modifications (Saido et al, 1995, 1996). This heterogeneity appears to affect catabolism and aggregation rates, resulting in structural changes in plaques. It has been reported repeatedly that most of Aβ in the brains of aged humans and in Down syndrome starts with pyroglutamate (pE) at position 3 and ends at position 42 (Aβ3pE-42) (Harigaya et al, 1995; Saido et al, 1995, 1996; Iwatsubo et al, 1996; Lemere et al, 1996; Kawarabayashi et al, 2001; Frost et al, 2013). In contrast, transgenic (Tg) mice over-expressing human APP with pathogenic mutations primarily accumulate Aβ1-40 and Aβ1-42 (Sturchler-Pierrat et al, 1997; Kawarabayashi et al, 2001). We reassessed this observation in the brains of AD patients quantitatively using a panel of antibodies capable of distinguishing among amino-terminal variants or among carboxyl-terminal variants of Aβ and verified that the predominant Aβ variant in the brains of AD patients is indeed Aβ3pE-42 (>40% of total Aβ) (Fig S1). There is, however, some discrepancy among different reports regarding the quantity of Aβ variants, which vary depending on the methods employed. For instance, the group led by Michel Goedert, who used MALDI-TOF mass spectrometry and LC-MS/MS to resolve the cryo-EM structure of Aβ42 filaments from human brains, recently indicated that Aβ3pE-42 was a minor variant in the brains of AD patients and App knock-in (KI) mice (Yang et al, 2022). This inconsistency can be accounted for by the unique physicochemical nature of Aβ3pE-42, which is seldom recovered from reversed-phase HPLC under normal conditions (Table S1). However, the use of a heated (50°C) basic solvent containing betaine and limited proteolysis using lysyl endopeptidase allows full recovery in HPLC and detection by mass spectrometry, respectively. These observations explain the relatively low estimation of Aβ3pE-42 levels in some prior studies as well (Glenner & Wong, 1984, 2012; Wong et al, 1985; Mori et al, 1992). Consistently, Güntert et al (2006) successfully detected Aβ3pE-42 in the human brain by mass spectrometry after sufficient proteolytic digestion. It is notable that a therapeutic antibody raised against this variant, donanemab, has

---

[1]Department of Genome-Based Drug Discovery and Leading Medical Research Core Unit, Graduate School of Biomedical Sciences, Nagasaki University, Nagasaki, Japan [2]Laboratory for Proteolytic Neuroscience, RIKEN Center for Brain Science, Saitama, Japan [3]Division of Neuroscience, Faculty of Biology, Medicine and Health, School of Biological Sciences, Faculty of Biology, Medicine and Health, School of Biological Sciences, The University of Manchester, Salford Royal Hospital, Salford, UK [4]Department of Neurocognitive Science, Institute of Brain Science, Nagoya City University Graduate School of Medical Sciences, Nagoya, Japan [5]Department of Neuroscience and Pathobiology, Research Institute of Environmental Medicine, Nagoya University, Nagoya, Japan [6]Department of Functional Brain Imaging, National Institutes for Quantum Science and Technology, Chiba, Japan

Correspondence: iwata-n@nagasaki-u.ac.jp; takaomi.saido@riken.jp

been shown to be effective in recent clinical trials (Demattos et al, 2012; Sims et al, 2023).

The selective deposition of this physiologically rare Aβ variant in the human brain appears to be attributed to its presumed metabolic stability and aggregation propensity because pyroglutamyl peptide is resistant to major aminopeptidases except for pyroglutamyl aminopeptidase (Mori et al, 1992). In accordance with this, an immediate aminopeptidase-sensitive precursor of Aβ3pE-42, Aβ3E-42, is a very minor component in AD brain (Fig S1) (Table S2). However, our observations of in vivo Aβ1-42 catabolism demonstrated that specific endoproteolysis mediated by neprilysin (NEP), but not aminopeptidase, is the major rate-limiting step with Aβ10-37 as a catabolic intermediate (Iwata et al, 2000). Thus, it is unclear about the mechanisms underlying generation of Aβ3pE-42. In addition, the presence of Aβ variants with different amino-terminal structures in the AD brains may be explained by the assumption that they have different life spans.

The present study aims to examine whether the amino-terminal structure of Aβx-42 influences its catabolic rate in vivo based on the diversity of its amino-terminal structure observed in the AD brain (Fig S1). The study also serves as an initial attempt to address the question "What determines the life spans of extracellular peptides, such as Aβs, in the brain?" For this purpose, we synthesized amino-terminal variants of Aβ internally radiolabeled with $^3$H at residues 2, 4, 9, and 17 and with $^{14}$C at 20, 29, 34, 40, and 42 (Fig S2) and analyzed their catabolic rates in the rat hippocampal tissue according to a previously reported method (Iwata et al, 2000). Furthermore, to elucidate possible pathways of Aβ3pE-40/42 generation, we analyzed amyloid pathologies in the brains of APP-Tg (Sturchler-Pierrat et al, 1997) and *App* KI mice (Saito et al, 2014) using various experimental paradigms, such as quantitative Western blot (WB) analysis, ELISA, mass spectrometry, and image analysis for immunohistochemistry and positron emission tomography (PET) of amyloid. We found that a deficiency of NEP, the major Aβ-degrading enzyme, up-regulated the compensatory pathway, where aminopeptidases, dipeptidyl peptidases, and glutaminyl-peptide cyclotransferase-like (QPCTL) are involved, resulting in the formation and relatively selective accumulation of Aβ3pE-42 in the brains of both APP-Tg and *App* KI mice.

## Results

### Catabolism of Aβx-42 variants in vivo

The synthesized variants included Aβ1D(L-Asp)-42, Aβ1rD(D-Asp)-42, Aβ1AcD(acetyl-Asp)-42, Aβ2A-42, Aβ3E-42, Aβ3pE-42, Aβ4F-42, and Aβ17L-42 (Fig S2). The results demonstrate that subtle differences in the amino-terminal structure have a profound influence on the way Aβ is catabolized (Figs 1A–F and S3, Table 1). First, Aβ3pE-42 was much more resistant to in vivo catabolism than Aβ1-42; most of it remained uncatabolized even 60 min after administration into the hippocampus, whereas Aβ1-42 was almost fully degraded within 30 min. This observation is consistent with the presumption attributing the selective deposition of Aβ3pE-42 in the human brain to its metabolic stability (Saido et al, 1995). In contrast, Aβ2A-42, Aβ3E-

42, and Aβ4F-42 were catabolized even more rapidly than Aβ1-42. These results clearly indicate the presence of structural determinants of life span in the amino-terminal sequence of Aβ. Again, these truncated Aβ peptides without pyroglutamate are consistently seen as minor components deposited in the senile human brain (Fig S1) (Saido et al, 1996). The catabolic stability of these Aβ variants therefore correlates well with their tendencies to be pathologically deposited in the human brain. The Aβ17L-42 (p3 fragment) appeared to have a longer in vivo half-life than Aβ1-42, but the difference was not statistically significant (Table 1).

These observations also indicate that the loss of the first two amino acid residues, DA, is not the cause of the metabolic stability of Aβ3pE-42. Loss of the α-amino group and R-configuration of the first amino acid did not affect the metabolic rates, because Aβ1AcD-42 and Aβ1rD-42 were degraded at a rate similar to that of Aβ1-42 (Table 1 and Fig S3). The specific structure of Aβ3pE-42 seems to generate an anti-catabolism signal, as discussed below. The consistently facilitated catabolism of Aβ2A-42, Aβ3E-42, and Aβ4F-42 also indicates that the amino-terminal aspartic residue of Aβ1-42 is involved in the regulation of the catabolic rate, generating an anti-catabolism signal that is milder than that of Aβ3pE-42. Because the rate-limiting step of Aβ1-42 degradation is catalyzed by NEP (Iwata et al, 2000, 2001), the conversion of Aβ1-42 to Aβ3pE-42 is likely to decelerate this process. The facilitation of catabolism by removal of aspartate from Aβ1-42, however, may possibly be accounted for by two distinct mechanisms: one is conversion to better substrates for NEP, and the other is participation of alternative catabolic pathway(s) with respect to Aβ2A-42, Aβ3E-42, and Aβ4F-42. In any case, NEP may contribute to catabolism, because a catabolic intermediate similar to Aβ10-37 detected during Aβ1-42 degradation (Iwata et al, 2000) was observed (Fig 1, green arrowheads). A detailed analysis of the peptide fragments generated during the in vivo catabolism is required to clarify this notion.

### Effect of *Mme* deficiency on N1D- and N3pE-positive amyloid plaque deposition in the APP-Tg mouse brain

We subsequently examined the effect of *Mme* (gene nomenclature of NEP) deficiency on deposition of amyloid plaques immunoreactive to various anti-Aβ antibodies including N1D and N3pE in the brains of APP-Tg mice (Fig 2A and B). *Mme* deficiency modestly increased N1D-positive amyloid plaques but induced a drastic increase in N3pE-positive plaques, particularly at and after 18 mo of age. Other immunostaining and thioflavin staining patterns were almost similar to that of N1D immunostaining. Large plaques were detected in the sections immunostained for N1D (Fig 2A) and plaque size distribution for both N1D- and N3pE-positive amyloid plaques. Interestingly, we found that plaque size was roughly divided into three clusters: smaller/cored plaques (≤300 $\mu$m$^2$), intermediate plaques (1,500–4,500 $\mu$m$^2$), and larger plaques (>20,000 $\mu$m$^2$) (Figs 2C and S4). The numbers of both N1D- and N3pE-positive amyloid plaques presented (Fig 2B) with a distinct size distribution: N3pE immunoreactivity tended to be present in smaller/cored plaques (≤300 $\mu$m$^2$) and intermediate plaques (1,500–4,500 $\mu$m$^2$), whereas N1D was present in larger plaques (>20,000 $\mu$m$^2$) in addition to the clusters of smaller/cored plaques and intermediate plaques. The relative increase in total N3pE-positive area by *Mme* deficiency was

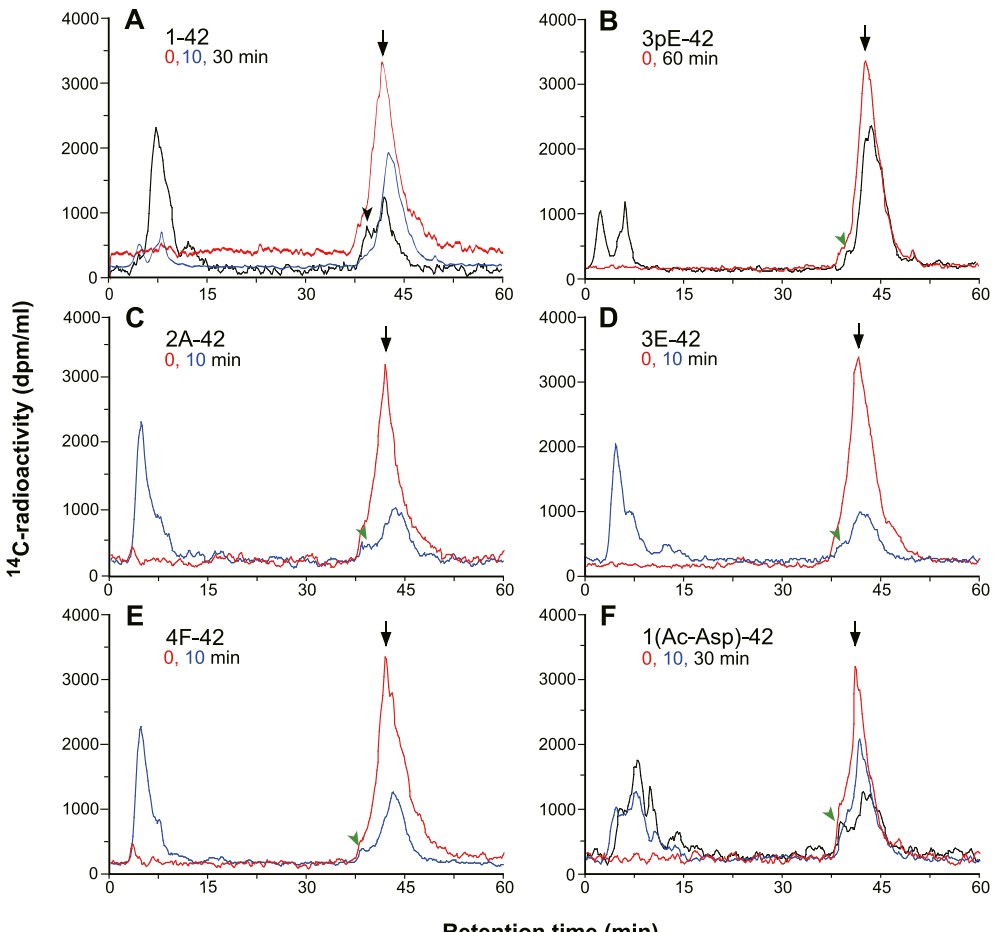

**Figure 1. Catabolism of Aβx-42 peptides in the hippocampus.**
**(A, B, C, D, E, F)** Variants of ³H/¹⁴C-Aβx-42 were subjected to in vivo degradation and subsequent analysis as previously described (Iwata et al, 2000). Briefly, 0.5 μg Aβx-42 peptide dissolved in phosphate-buffered saline was unilaterally injected into the CA1 sector of the rat hippocampus and subjected to catabolism until the animals were euthanized by decapitation at the indicated time points. There was no contamination of the injection area by plasma or cerebrospinal fluid as immunohistochemically confirmed using anti-rat IgG. After extraction, the products were analyzed by reversed-phase HPLC connected to a flow scintillation monitor. The HPLC profiles in a ¹⁴C mode at time 0 (red), 10 (blue), and 30/60 (black) minutes after the in vivo injection are shown. The elution profiles in ¹⁴C and ³H modes were essentially identical. The major peaks at time 0 indicated by the arrows correspond to the intact substrates. The black arrowhead shows Aβ10–37 in (A) (Aβ1-42), and green arrowheads show possible intermediates in other panels. Digitized data were used to calculate the in vivo half-lives of the peptides as shown in Fig S2 and Table 1.

**Table 1. In vivo half-lives of Aβx-42.**

| X (amino-terminal residue) | Half-life (min)[a] | P-value against 1(Asp)[b] | |
|---|---|---|---|
| 1(Asp) | 18.08 | — | |
| 2(Ala) | 10.77 | 0.02 | significant |
| 3(Glu) | 12.51 | 0.016 | significant |
| 4(Phe) | 10.51 | 0.005 | significant |
| 3(pyroGlu) | 90.25 | 0.005 | significant |
| 17(Leu) | 34.69 | 0.081 | |
| 1(Ac-Asp) | 19.12 | 0.816 | |
| 1(D-Asp) | 20.53 | 0.662 | |

[a]Calculated using a formula of the exponential approximation curve when the y-intercept was 0.
[b]Analyzed by two-way ANOVA, followed by a *post hoc* test.

significantly larger than that of N1D-positive area (Fig 2D). Given that NEP expression invariably declines with age (Iwata et al, 2002; Wang et al, 2003; Russo et al, 2005; Hellstrom-Lindahl et al, 2008), it is notable that even heterozygous *Mme* deficiency, presenting an ~50% reduction in NEP activity (Iwata et al, 2001), thus resulted in a

relative increase in the ratio of Aβ3pE-40/42 to Aβ1-40/42 (Fig 2D). We then analyzed the detergent-insoluble/formic acid–soluble fractions of these mouse brains by mass spectrometry after the digestion of the samples by lysyl endopeptidase (Figs 3A and B and S5 and Table S3). We detected not only Aβ1-16 but also Aβ3pE-16 in the brains of *Mme*⁻/⁻APP-Tg mice (Fig 3B). Interestingly, this fraction contained trace quantities of Aβ2-16 and Aβ3-16, which are precursors of Aβ3pE-x. Next, we further analyzed amounts of Aβs present in the soluble fraction and accumulated in the insoluble fraction of the brains of APP-Tg mice using Western blot (WB) analysis (Figs 4A–C and S6) and ELISA (Fig 4D), which showed that changes in amounts of Aβ immunoreactive to the anti-Aβ antibodies basically showed similar patterns as that obtained using immunohistochemistry: *Mme* deficiency prominently increased the amounts of Aβ3pE-x from middle age, but a difference in the amount of Aβ1-x between the two genotypes was not observed with the progression of age. In contrast, the amount of Aβ3pE-x in both genotypes kept increasing until 24 mo of age (Fig 4B and C), resulting in an increased ratio of Aβ3pE-40/42 to Aβ1-40/42 (Fig 4D, rightmost graph). Notably, detectable amounts of Aβ3pE-40/42 were present both in the soluble and in the insoluble fractions (the order level of pmol/g), although they were considerably less than that of Aβ1-40/42. Colocalization of Aβ3pE-42 with cored

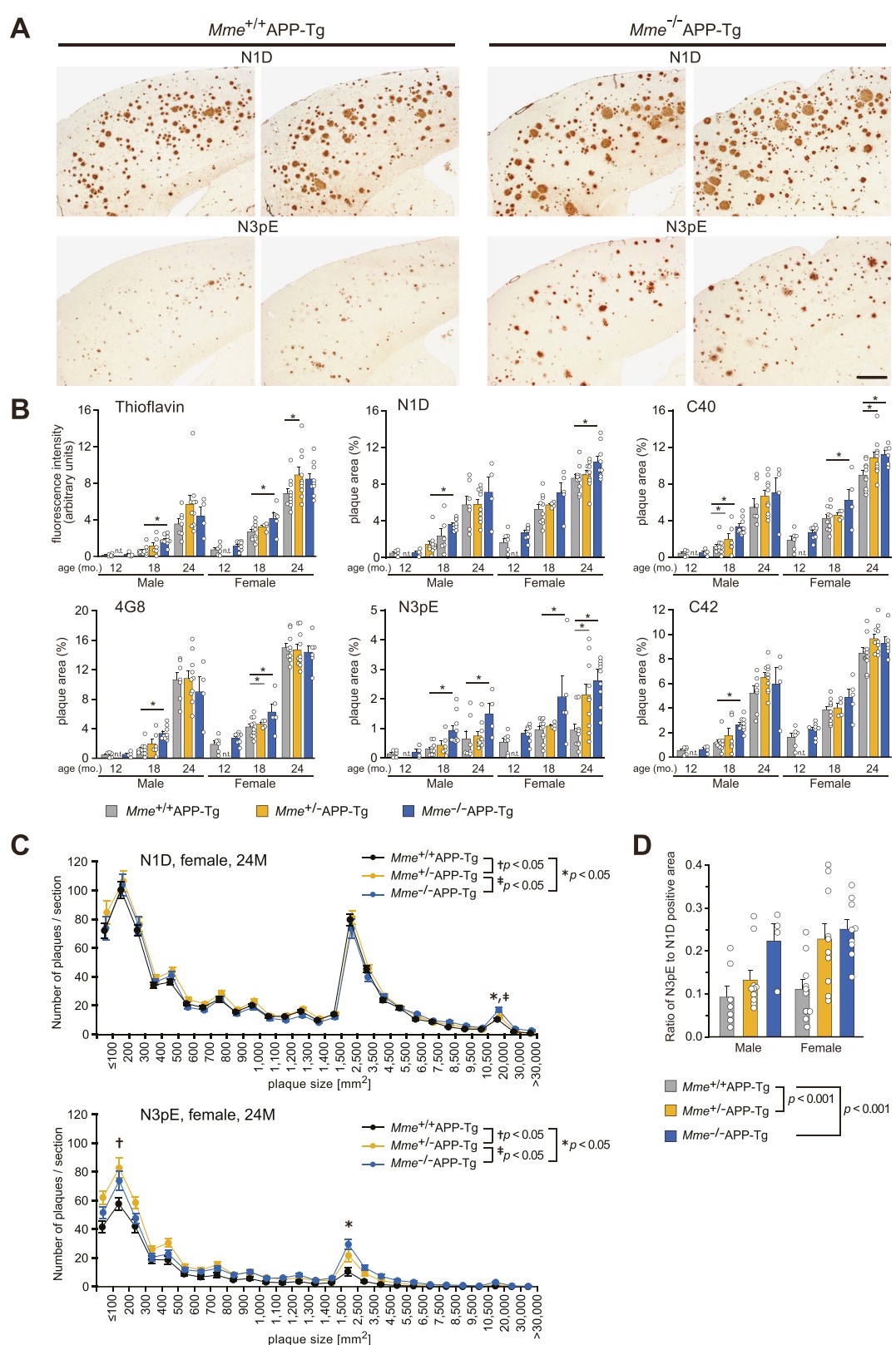

**Figure 2. Deposition of Aβ1-42 and Aβ3pE-42 in the brains of aged *Mme*-deficient APP-Tg mice.**
**(A)** Immunohistochemical staining of the brains of 24-mo-old *Mme*⁺/⁺APP-Tg and *Mme*⁻/⁻APP-Tg mice using anti-Aβ antibodies, N1D and N3pE. Immunostained sections from two mice in each case are shown. Scale bar, 500 μm. **(B)** Aβ deposits in brain sections from *Mme*⁺/⁺APP-Tg and *Mme*⁻/⁻APP-Tg mice (aged 12, 18, and 24 mo) were

plaques in the $Mme^{-/-}$APP-Tg mouse brain was examined using Pittsburgh compound B (PiB), which is known to selectively bind to cored plaques (Maeda et al, 2007; Ikonomovic et al, 2008) (Fig 5). Aβ3pE-42–positive signals corresponding to PiB-binding sites were detected preferentially in the core of amyloid plaques, whereas Aβ1-42–positive signals were observed in larger amyloid plaques (Fig 5A) and the PiB binding was increased in $Mme^{-/-}$APP-Tg mice in a manner colocalizing with N3pE-positive cored plaques in a significant way (Fig 5B, left panel, and Fig 5C). PET analysis also showed an increase of in vivo binding of PiB in the $Mme^{-/-}$APP-Tg mice in an age-dependent manner (Fig 5B, right panel, and Fig 5D). Furthermore, we found that Aβ3pE-40/42, but not Aβ1-40/42, colocalized well with apolipoprotein E (ApoE), which is known to be involved in Aβ aggregation (Nilsson et al, 2004), in the cores of amyloid plaques (Fig S7A). Intensities of ApoE-positive signals in the amyloid plaques and levels of ApoE and α1-antichymotrypsin in the soluble fraction were significantly elevated by the $Mme$ deficiency in the 24-mo-old APP-Tg mice (Fig S7B and C). These observations consistently indicated that $Mme$ deficiency increases the cored plaque–associated Aβ3pE-42.

### Generation and analysis of $App^{NL-(\Delta DA)-F}$ and $App^{NL-(\Delta DA)-Q-F}$ knock-in mouse lines

We next aimed to reproduce Aβ3pE-42 pathology in mouse models, for which two knock-in mouse lines were generated: $App^{NL-(\Delta DA)-F}$ and $App^{NL-(\Delta DA)-Q-F}$ (Fig 6A). This strategy was based on our previous experimental results showing that primary neurons expressing APP cDNA with the first two amino acid residues of Aβ deleted (NL[ΔDA]E) produced Aβ3E-40/42 and that those in which E had been replaced by Q (NL[ΔDA]Q) produced Aβ3pE-40/42 (Shirotani et al, 2002). First, using the $App^{NL-F}$ line of $App$ KI mice we confirmed the effects of $Mme$ deficiency as obtained with APP-Tg mice. Incidentally, $Mme$ deficiency not only accelerated the deposition of both Aβ1-42 and Aβ3pE-42 in the $App^{NL-F}$ line, but also increased the ratio of Aβ3pE-42 to Aβ1-42 (Fig 6B). When we analyzed new lines of $App$

KI mice, we were indeed able to detect a trace amount (~1 fmol/g) of Aβ3E-42 only in the brains of $App^{NL-(\Delta DA)-Q-F}$ mice at 2 mo of age (Fig S8). Despite our expectation, neither the $App^{NL-(\Delta DA)-F}$ nor the $App^{NL-(\Delta DA)-Q-F}$ lines exhibited any visible Aβ pathology at all, even after aging (i.e., 18 mo or more) (Fig 6C). Notably, biochemical quantification indicated that the $App^{NL-F}$ line deposited more Aβ3pE-42 than the $App^{NL-(\Delta DA)-F}$ and $App^{NL-(\Delta DA)-Q-F}$ lines (Fig 6D), consistently indicating that Aβ3E-42 and Aβ3Q-42 are much more short-lived than Aβ1-42 after production in vivo.

In any case, both histochemical and biochemical analyses of APP-Tg (Figs 2, 4, and 5) and $App^{NL-F}$ line of $App$ KI mice (Figs 6B and S9A) consistently indicated that deposition of Aβ3pE-42 takes place much later than that of Aβ1-42. The biochemical quantity of Aβ3pE-42 accounts for only 0.1% of total Aβ42 in the insoluble fraction of 24-mo-old $App^{NL-F}$ mouse brains (Fig S9B and C).

### Changes in expression levels of exopeptidases and glutaminyl-peptide cyclotransferases involved in AβN3pE generation by $Mme$ deficiency

We assumed that a potential pathway where aminopeptidases and dipeptidyl peptidases may be involved (Fig S10) could compensate for physiological Aβ degradation decreased by $Mme$ deficiency. Our findings show that aminopeptidase A (APN), aminopeptidase N (APN), and dipeptidyl peptidase 4 (DPP4), which were immuno-purified from the membrane fraction of mouse brains, displayed Aβ-degrading activity (Fig 7A). Among them, DPP4 exhibited the highest activity. Protein contents were in the order of APN > APA > DPP4, but DPP4 had the highest specific activity (Fig 7C), suggesting that it plays a significant role in truncating amino-terminal amino acids (DA). Our results showed that the expression levels of APN, DPP4, and glutaminyl-peptide cyclotransferase-like (QPCTL) were up-regulated by $Mme$ deficiency, whereas the expression level of thyrotropin-releasing hormone–degrading enzyme, a pyroglutamyl aminopeptidase, was decreased (Fig 7D and E).

---

stained with thioflavin or immunostained with amino-terminal specific antibodies for Aβ (N1D and N3pE), anti-pan Aβ antibody (4G8), and carboxyl-terminal specific antibodies for Aβ (C40 and C42), and then quantified as described in the Materials and Methods section. Amyloid load is expressed as a fluorescence intensity in the measured area or a percent of the measured area. Each column with a bar represents the mean ± SEM. Multiple comparisons were done by a one-way ANOVA, followed by a *post hoc* test as described in the "Materials and Methods" section. The numbers of analyzed animals were as follows: 12 mo old, 5 male and 6 female $Mme^{+/+}$APP-Tg, and 4 male and 8 female $Mme^{-/-}$APP-Tg; 18 mo old, 10 male and 10 female $Mme^{+/+}$APP-Tg, 6 male and 4 female $Mme^{+/-}$APP-Tg, and 9 male and 5 female $Mme^{-/-}$APP-Tg; and 24 mo old, 7 male and 10 female $Mme^{+/+}$APP-Tg, 10 male and 10 female $Mme^{+/-}$APP-Tg, and 4 male and 9 female $Mme^{-/-}$APP-Tg mice. *$P$ < 0.05, significantly different from $Mme^{+/+}$APP-Tg mice in the same ages. n.t., not tested. **(C)** Aβ plaque sizes in brain sections from $Mme^{+/+}$APP-Tg, $Mme^{+/-}$APP-Tg, and $Mme^{-/-}$APP-Tg mice (female, 24 mo old) were analyzed using MetaMorph image analysis software. Two or three sections from one individual (a total of 172 sections) were analyzed, and data were averaged. The area per section analyzed was 13.8 ± 0.058 mm². The numbers of analyzed animals were as follows: 11 female $Mme^{+/+}$APP-Tg, 10 female $Mme^{+/-}$APP-Tg, and 9 female $Mme^{-/-}$APP-Tg mice. Each point with a bar represents the mean ± SEM. Where the SEMs are not shown, they are smaller than the symbols. Differences in the pattern of plaque size distribution between N1D- and N3pE-positive plaques in all cohorts of mice were analyzed by repeated-measures two-way ANOVA, which revealed significant main effects of plaque size ($F_{(26, 754)}$ = 501.535; $P$ < 0.001) and amino-terminal structure ($F_{(1, 754)}$ = 343.330; $P$ < 0.001) and a significant interaction ($F_{(26, 754)}$ = 75.525; $P$ < 0.001). Interactions between the amino-terminal structure and genotype for particular plaque sizes (100–200, 1,500–2,500, and 10,500–20,000 μm²) were also analyzed by repeated-measures two-way ANOVA, followed by a *post hoc* test. $F$ and $P$-values are as follows: $F_{(2,27)}$ = 2.291 ($P$ = 0.121), $F_{(2,27)}$ = 7.925 ($P$ = 0.002), and $F_{(2,27)}$ = 3.133 ($P$ = 0.060), respectively. The asterisk, dagger, and double dagger indicate a significant difference with $P$-values less than 0.05. Data from APP-Tg mice (male, 24 mo old, and female, 18 mo old) are shown in Fig S4. **(D)** Ratios of N3pE- to N1D-positive areas in $Mme^{+/+}$APP-Tg, $Mme^{+/-}$APP-Tg, and $Mme^{-/-}$APP-Tg mouse brains (male and female, 24 mo old) were compared. Immunostaining for N3pE and N1D was carried out using two to three sets of serial brain sections from the individual, and the average was used as a determinant. Each column with a bar represents the mean ± SEM. The make-up of the animal group (number, males, females, etc.) was the same as that shown in Fig 2B. Two-way ANOVA (gender, genotype) showed a significant main effect of the $Mme$ genotype ($F_{(2, 44)}$ = 9.434; $P$ < 0.001). Ratios of N3pE- to N1D-positive areas in $Mme^{+/+}$APP-Tg were significantly different from other genotypes (*$P$ < 0.001).

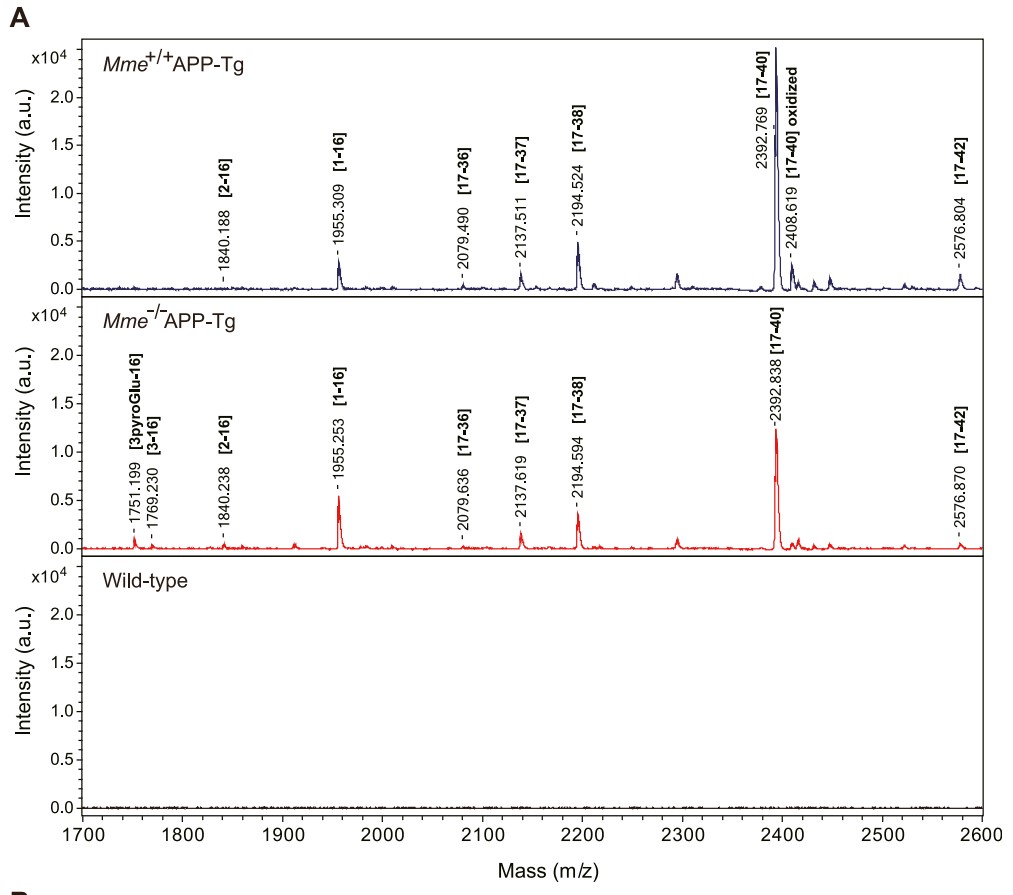

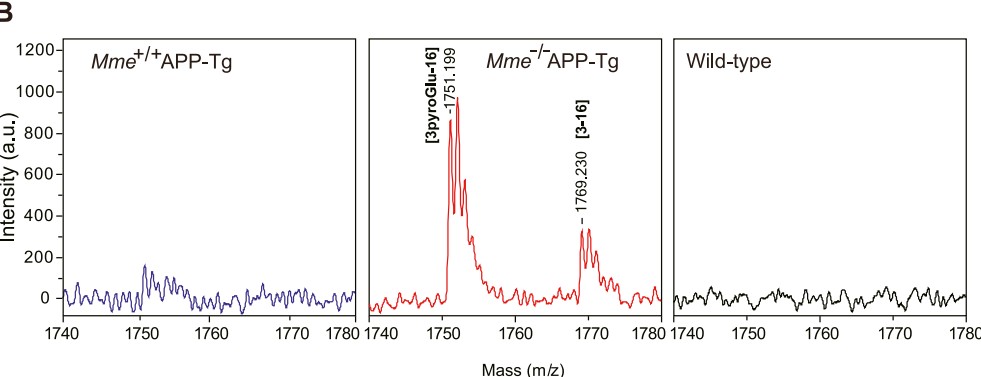

**Figure 3. Mass spectrometric profiles from detergent-insoluble/formic acid–soluble fractions of brains from aged APP-Tg mice.**
**(A)** Monoisotopic mass of Aβ variants in the detergent-insoluble/formic acid–soluble fractions from 24-mo-old APP transgenic and non-transgenic brains was determined after the digestion of lysyl endopeptidase, because Aβ with 3pE at the amino-terminal shows poor ionization properties. **(A, B)** m/z range between 1,740 and 1,780 in panel (A) is shown at a higher magnification. The mass signals without annotation were not related to Aβ. Mass spectrometric profiles in ranges of m/z 1,480–1,940 and m/z 4,000–4,600 are shown in Fig S5.

## Discussion

An unresolved issue in understanding the mechanism of Aβ deposition in the human brain, an upstream event triggering the AD cascade (Selkoe & Hardy, 2016), is the difference in the primary structure of Aβ between the pathologically deposited and physiologically secreted forms (Saido et al, 1995, 1996; Saido & Iwata, 2006). Gravina et al demonstrated using end-specific antibodies that most Aβ in the AD brain is amino-terminally truncated and that Aβ1-42 accounts for only 10–20% of total Aβx-42 (Gravina et al, 1995), whereas Aβ1-40 and Aβ1-42 are the major variants secreted by cells (Selkoe & Hardy, 2016). The actual amount of a physiologically secreted form, Aβ1(L-Asp)-42, in the AD brain was even smaller (less than 5%) (Fig S1). This specific form accounts for >40% of total Aβx-42 as quantitated against varying amounts of synthetic Aβ peptides, in agreement with Kuo et al (1997); Russo et al (1997). Thus, among the structural variants known to be present in AD brains, Aβ3pE-42 is the most abundant in both early- and late-onset cases. The next most abundant Aβ variant after Aβ3pE-42 was Aβ1iD-42, which was estimated to be 26% of the total Aβ (Fig S1). Moro et al (2018) observed that the amount of Aβ3pE-42 was correlated well with phosphor-tau load in AD and age-matched control brains rather than that of Aβ1iD-42; thus, deposition of Aβ3pE-42 appears to be associated with AD pathogenesis.

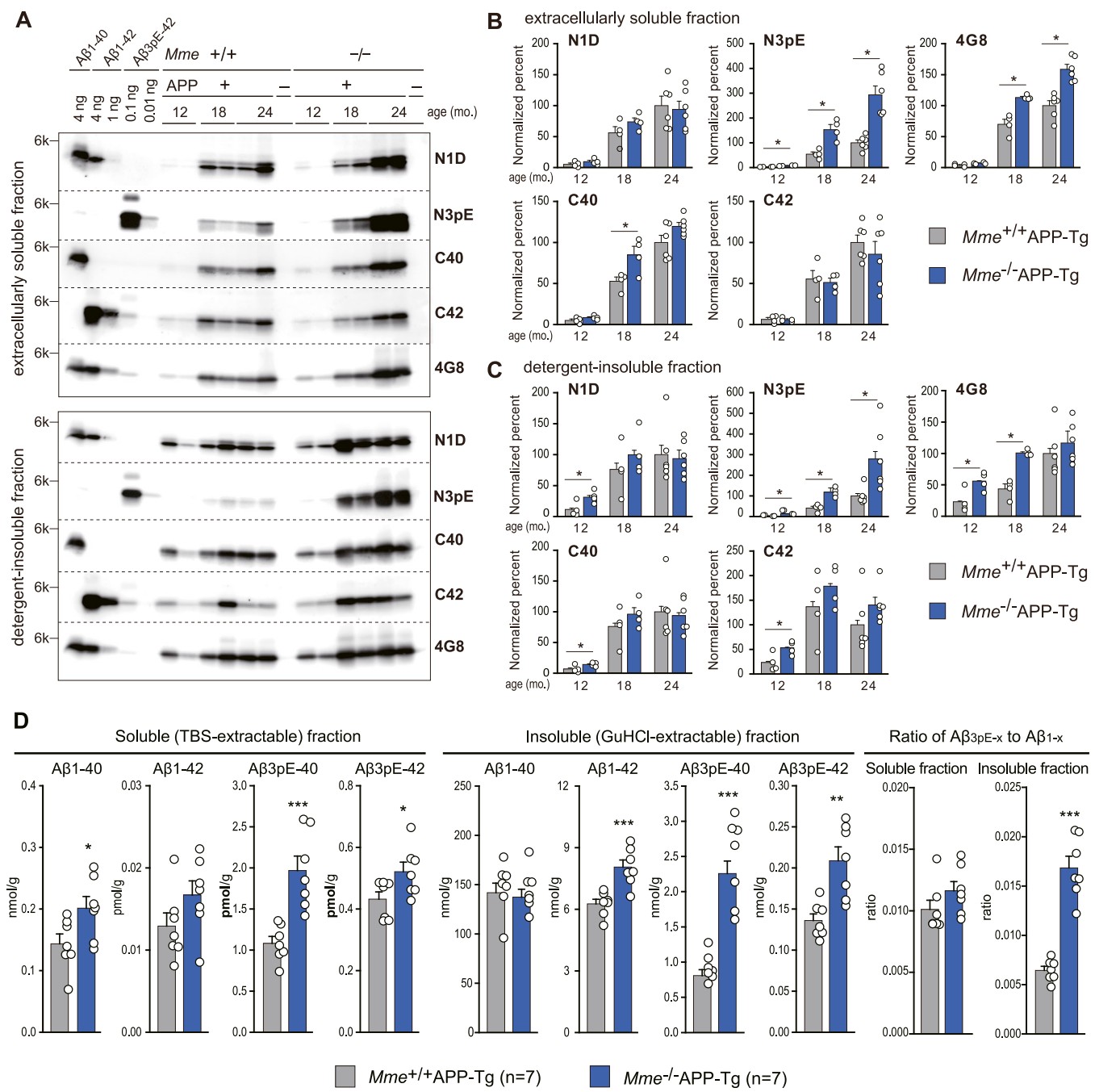

**Figure 4. Diverse effects of *Mme* deficiency on extracellularly soluble and insoluble levels of Aβ variants in aged *Mme*-deficient APP-Tg mice.**
**(A)** Brain tissue fractions (upper panel: extracellularly soluble fraction, 0.5 μg protein; lower panel: detergent-insoluble/formic acid–extractable fraction, 20 ng protein) of non-transgenic and APP transgenic mice with or without the *Mme* gene (12, 18, and 24 mo old) were subjected to Western blot analyses using amino-terminal specific antibodies for Aβ (N1D and N3pE), anti-pan Aβ antibody (4G8), and carboxyl-terminal specific antibodies for Aβ (C40 and C42). **(B, C)** Extracellularly soluble fraction and (C) detergent-insoluble/formic acid–extractable fraction intensities of immunoreactive bands on blots shown in Fig S6 were quantified as described in experimental procedures. The intensities were normalized against the data from 24-mo-old *Mme*^+/+^APP-Tg mice. **(B, C)** Amounts of immunoreactive Aβ variants in 24-mo-old *Mme*^+/+^APP-Tg mouse brains to N1D, N3pE, 4G8, C40, and C42 are 15.80, 0.028, 10.12, 8.18, and 1.70 ng/μg protein in (B), and 241.3, 1.150, 895.7, 461.6, and 91.7 ng/μg protein in (C), respectively. Each column with a bar represents the mean ± SEM of 4–6 female mice. *$P < 0.05$, significantly different from *Mme*^+/+^APP-Tg mice at the same ages. Two-way ANOVA revealed significant interactions between *Mme* deficiency and ages on N3pE levels in soluble ($F_{(2,22)}$ = 14.612; $P < 0.001$) and in the detergent-insoluble/formic acid–extractable ($F_{(2,22)}$ = 1.193; $P < 0.05$) fractions. **(D)** Levels of Aβ1-40, Aβ1-42, Aβ3pE-40, and Aβ3pE-42 in soluble (TBS-extractable) and insoluble (GuHCl-extractable) fractions extracted from the brains of 24-mo-old female *Mme*^+/+^APP-Tg and female *Mme*^−/−^APP-Tg mice were determined using sandwich ELISA. Percentages of Aβ1-x and Aβ3pE-x to total Aβ (sum of Aβs from both the soluble and insoluble fractions) and ratios of Aβ3pE-x to Aβ1-x (sum of both Aβx-40 and Aβx-42) in soluble and insoluble fractions are shown in the right panels. Each column with a bar represents the mean ± SEM of seven mice. Significant differences between two groups were determined using a *t* test or a Mann–Whitney *U* test (*$P < 0.05$, **$P < 0.005$, and ***$P < 0.001$).

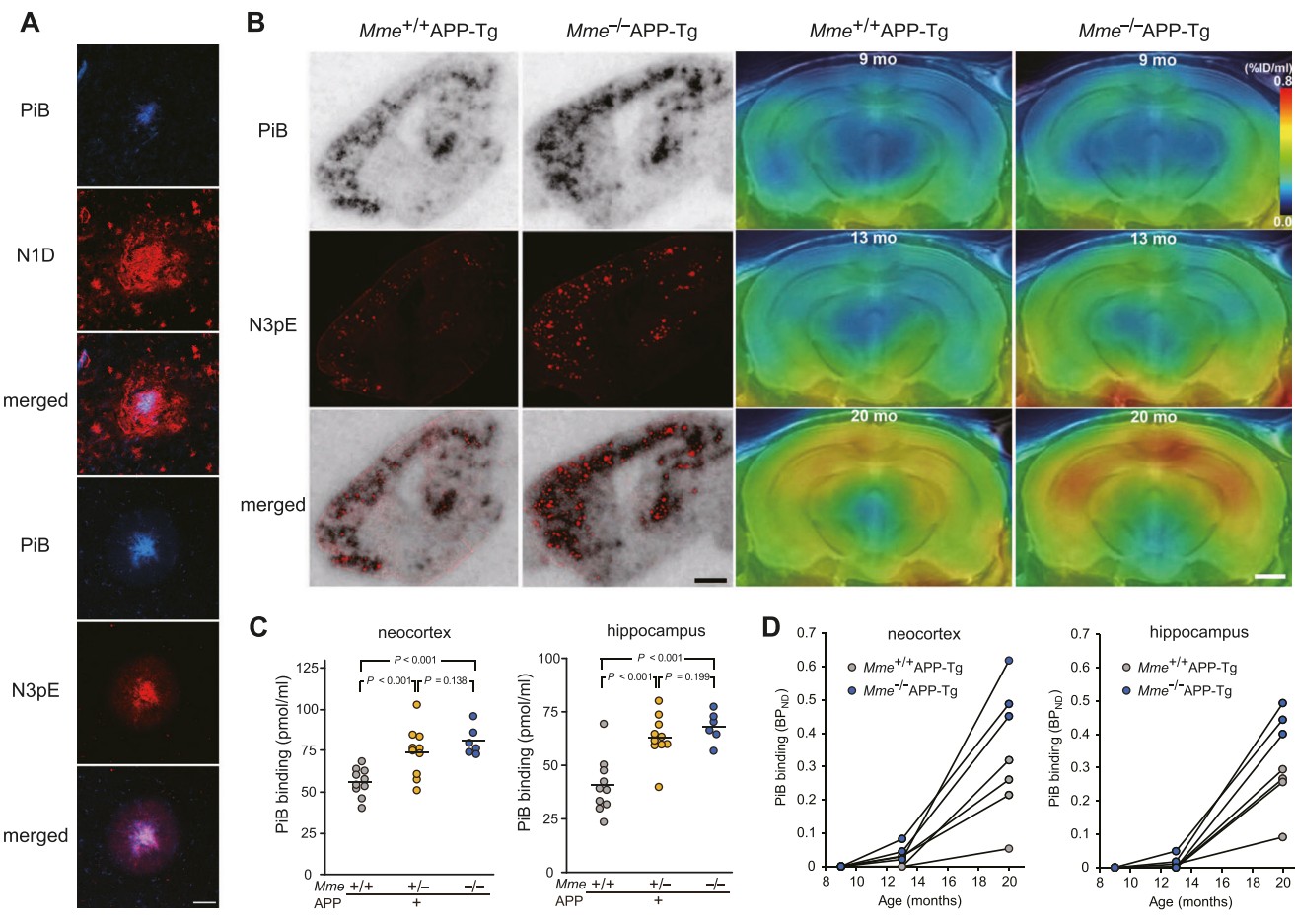

**Figure 5. Association of in vitro and in vivo radiolabeling of amyloid with [11C]PiB and Aβ3pE-42.**
**(A)** Confocal fluorescence microscopic images of amyloid plaques in 24-mo-old APP-Tg mice doubly stained with PiB/N1D (top 3 panels) and PiB/N3pE (bottom 3 panels). Scale bar, 50 μm. **(B)** Brain sections from 24-mo-old $Mme^{+/+}$APP-Tg and $Mme^{-/-}$APP-Tg mice were subjected to autoradiography with [11C]PiB (*left panel*), and thereafter immunostained with polyclonal anti-Aβ3pE antibody (*right panel*). Colocalization of radiolabeling with Aβ3pE deposition is shown in the lower panels. Scale bar, 1.0 mm. **(C)** Intensities of [11C]PiB signals (normalized by the intensity of non-specific cerebellar labeling) in the neocortex and hippocampus of 24-mo-old APP-Tg were significantly elevated and inversely correlated with the *Mme* gene dose. Open symbols show individual values obtained by the in vitro autoradiography of brain sections. A solid bar represents the mean value in each group. The following genotypes were tested: $Mme^{+/+}$APP-Tg (10 females), $Mme^{+/-}$APP-Tg (10 females), and $Mme^{-/-}$APP-Tg (6 females). **(D)** Longitudinal changes of in vivo [11C]PiB retentions estimated as non-displaceable binding potential ($BP_{ND}$) in the neocortex (left) and hippocampus (right) of $Mme^{+/+}$APP-Tg (open circles; n = 4) and $Mme^{-/-}$APP-Tg (closed circles; n = 3) mice at 9 (top), 13 (middle), and 20 mo of age. Data from the same individuals are connected by solid lines. There were significant main effects of age ($F_{(2,4)}$ = 70.9 and 118.9; $P$ < 0.0001 in the neocortex and hippocampus, respectively) and genotype ($F_{(1,5)}$ = 18.7; $P$ < 0.01; and $F_{(1,5)}$ = 15.9; $P$ < 0.05 in the neocortex and hippocampus, respectively) detected by repeated-measures two-way ANOVA.

Aβ3pE-42 has been shown to exhibit greater neurotoxicity and oligomerization properties than Aβ1-40 and Aβ1-42 (He & Barrow, 1999; Jawhar et al, 2011; Nussbaum et al, 2012; Frost et al, 2013; Wulff et al, 2016; Dunkelmann et al, 2018a, 2018b). Conversion of Aβ1-42 to Aβ3pE-42 results in the loss of one positive and two negative charges at the N-terminus of Aβ (Saido et al, 1995), and may thus account for its unique physical, chemical, and biological characteristics. The generation of this peculiar Aβ variant may thus play a major pathogenic role in AD development and resulted in the discovery of donanemab. The mechanism by which Aβ3pE-42 is generated is likely to be mediated by amino-terminal truncation of Aβ1-42 by exopeptidase(s), such as aminopeptidases or dipeptidyl peptidases, followed by cyclization of the amino-terminal glutamate residue in Aβ3pE-42 (Fig S10). The cyclization may be spontaneous or enzymatic (Cynis et al, 2006; Antonyan et al, 2018), but to

our knowledge, substantial conversion of AβE-42 to Aβ3pE-42 under physiological conditions has never been demonstrated even in vitro (Shirotani et al, 2002).

Our findings suggest the following hypothetical scenario for pathological Aβ3pE-42 deposition in the human brain, as the conversion of Aβ3E-42 to this specific form can be achieved by a single chemical step involving dehydration/cyclization of the amino-terminal glutamate residue (Fig S10). Under normal conditions, the NEP-dependent pathway, which is constitutively active, is sufficient to catabolize Aβ1-42 without amino-terminal proteolysis to produce Aβ2A-42, Aβ3E-42, and Aβ4F-42. This conclusion was reached because aminopeptidase inhibitors did not block the in vivo degradation of radiolabeled Aβ1-42 (Iwata et al, 2000), and because Aβ1(Ac-Asp)-42 and Aβ1(D-Asp)-42, both of which are resistant to mammalian aminopeptidase activity

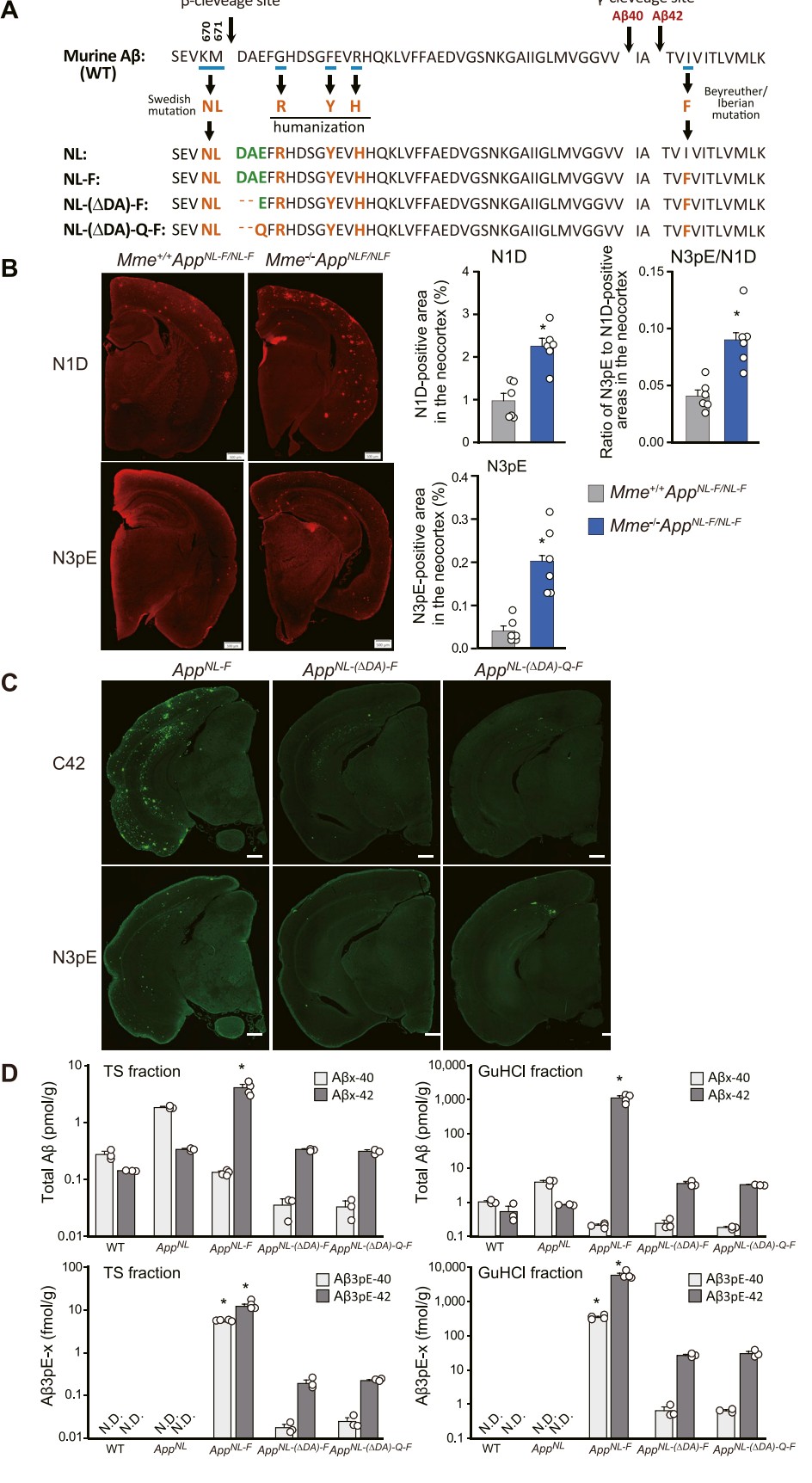

**Figure 6. Generation of *App^NL-(ΔDA)-F* and *App^NL-(ΔDA)-Q-F* KI mice and their Aβ pathology.**
**(A)** Amino acid sequences of humanized Aβ with mutations in several lines of App KI mice. In this study, we generated two additional lines of *App* KI mice: in the *App^NL-(ΔDA)-F* KI mice, the first two amino acids of Aβ (DA) are deleted. In the *App^NL-(ΔDA)-Q-F* KI mice, the third amino acid of Aβ (E) in *App^NL-(ΔDA)-F* is converted to Q. These experimental designs were made based on our previous observation, showing that cortical neurons expressing APP cDNAs with the NL-(ΔDA)-Q mutation, but not with NL or NL(ΔDA) mutations, produced Aβ3pE-40/42 (Shirotani et al, 2002). **(B)** Immunohistochemical staining of the brains of 12-mo-old *Mme^+/+App^NL-F* and *Mme^-/-App^NL-F* KI mice using anti-Aβ antibodies, N1D and N3pE. Scale bar, 500 μm. Amyloid load is expressed as a fluorescence intensity in the measured area or a percent of the measured area. Each column with a bar represents the mean ± SEM of 3 female and 3 male mice. An asterisk shows a significant difference with a *P*-value less than 0.05. **(C)** Immunohistochemical staining of the brains of 18-mo-old *App^NL-(ΔD)-F* and *App^NL-(ΔDA)-Q-F* KI mice with/without *Mme* using anti-Aβ antibodies, C42 and N3pE. Scale bar, 500 μm. **(D)** Levels of Aβx-40, Aβx-42, Aβ3pE-40, and Aβ3pE-42 in soluble (TBS-extractable) and insoluble (GuHCl-extractable) fractions extracted from the brains of several lines of 18-mo-old App KI mice were determined by sandwich ELISA. Each column with a bar represents the mean ± SEM of three to four mice. TS-Aβ40/42: two-way ANOVA (genotype and amino-terminal of Aβ) showed a significant main effect of the genotype ($F_{(4, 22)}$ = 33.801; $P < 0.001$). GuHCl-Aβ40/42: two-way ANOVA (genotype and amino-terminal of Aβ) showed a significant main effect of the genotype ($F_{(4, 22)}$ = 34.534; $P < 0.001$). TS-Aβ3pE40/42: two-way ANOVA (genotype and amino-terminal of Aβ) showed a significant main effect of the genotype ($F_{(4, 22)}$ = 61.273; $P < 0.001$). GuHCl-Aβ3pE40/42: two-way ANOVA (genotype and amino-terminal of Aβ) showed a significant main effect of the genotype ($F_{(4, 22)}$ = 46.041; $P < 0.001$). Levels of Aβx-40(42) or Aβ3pE-40(42) in *App^NL-F* were significantly different from other genotypes in each fraction (*$P < 0.001$).

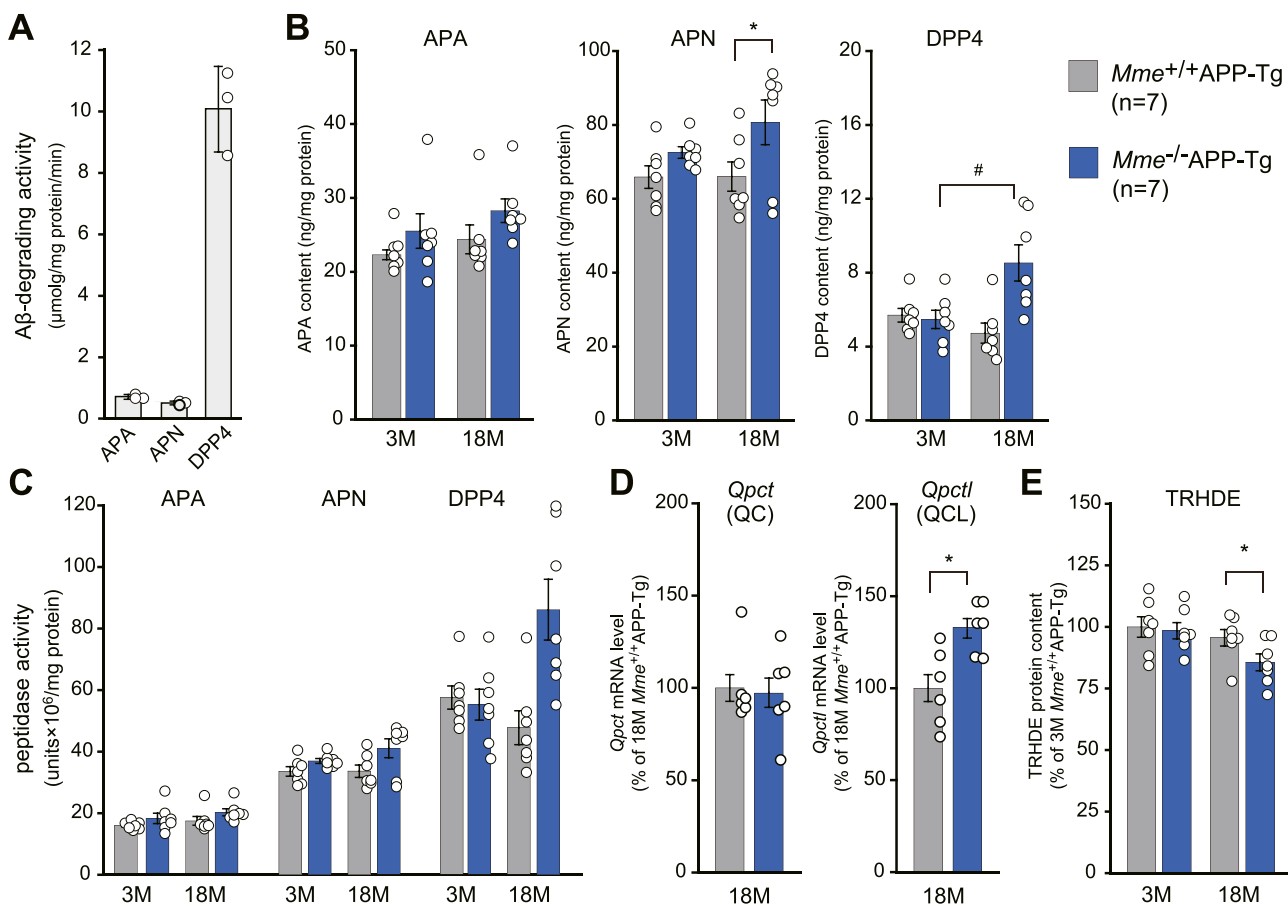

**Figure 7. Changes in expression levels of exopeptidases and glutaminyl-peptide cyclotransferases involved in Aβ3pE-x generation by *Mme* deficiency.**
**(A)** Aβ1-40–degrading activities by immunopurified APA, APN, and DPP4 from mouse brains. Aβ1-40 (2 mg) was incubated with each peptidase at 37°C for 5 h, and the amount of intact Aβ remaining was quantified using an HPLC system as described in the Materials and Methods section. Each column represents the mean ± SD of three assays. **(B)** Protein contents of APA, APN, and DPP4 in the brain membrane fractions of *Mme*+/+APP-Tg and *Mme*−/−APP-Tg mice at different ages (3 and 18 mo old) were determined by Western blot analysis with a specific antibody against each peptidase. Each column represents the mean ± SEM of seven mice. An asterisk and a hash show a significant difference at *P* < 0.05. **(A, B, C)** Specific activities for Aβ1-40 degradation by each peptidase were calculated based on the data from (A, B). **(D)** Levels of glutaminyl-peptide cyclotransferase (*Qpct*) and glutaminyl-peptide cyclotransferase-like (*Qpctl*) mRNAs in the cerebral cortices of *Mme*+/+APP-Tg and *Mme*−/−APP-Tg mice at 18 mo of age were determined by qRT-PCR. Each column represents the mean ± SEM of seven mice. Statistical analysis was carried out by a *t* test with ΔCt values. Asterisks show a significant difference at *P* < 0.05. **(E)** Protein contents of TRHDE in the brain membrane fractions of *Mme*+/+APP-Tg and *Mme*−/−APP-Tg mice at different ages (3 and 18 mo old) were determined by Western blot analysis with a specific antibody. Each column represents the mean ± SEM of seven mice. An asterisk shows a significant difference at *P* < 0.05.

(Harigaya et al, 1995), underwent in vivo degradation in a manner similar to that of Aβ1-42 (Table 1). However, under aberrant conditions, such as a significant reduction in NEP activity presumably caused by brain aging processes and AD development, the NEP-independent pathway(s) may be opened and Aβ1-42 may be processed by aminopeptidases or dipeptidyl peptidases, which are present in the extracellular milieu in the brain (Mentlein, 2004; Banegas et al, 2006; Hui, 2007; Khosla et al, 2022) (Figs 7 and S10). In support of this, these exopeptidases were abnormally up-regulated in an age-dependent manner to compensate for reduced NEP-mediated endoproteolysis. Although most of the amino-terminally truncated Aβx-42 underwent further proteolysis, a small portion of Aβ3E-42 could be converted to the catabolism-resistant form by QPCT or QPCTL (Fig S10), increasing the probability of pathological Aβ deposition.

Interestingly, we detected Aβ3pE-16 and trace quantities of Aβ2A-16 and Aβ3E-16 in *Mme*−/−APP-Tg mouse brain (Fig 3B). Furthermore, *Mme* deficiency increased not only the histochemical quantity of amyloid plaques but also the biochemical quantity of Aβ3pE-42 in the soluble fraction of APP-Tg brains as analyzed by WB and ELISA (Figs 4 and S6). The presence of Aβ3pE-42 in a soluble form even before plaque formation in the human brain (Russo et al, 1997) indicates the presence of a dynamic equilibrium between the liquid and solid phases. These observations imply that the N-terminal truncation of Aβ2A-x and conversion from Aβ3E-x to Aβ3pE-x may take place both in a solid state such as the core of amyloid plaques and in a liquid–solid transition state, such as formation of amyloid plaques from soluble Aβ monomers and oligomers. Aβ3pE-42–positive signals corresponding to PiB-binding sites were detected preferentially in the core of amyloid plaques, whereas Aβ1-42–positive signals were observed in larger amyloid

plaques (Fig 5A). If Aβ3pE-42 generation occurred only in the solid state, it is difficult to account for this morphological difference.

The relative amount of Aβ3pE-42 per total Aβ in the APP-Tg and *App* KI mouse models was less than 1% (Figs 4D and E and S9C) and did not recapitulate the massive accumulation of Aβ3pE-42 in the human brain (Fig S1), even if Aβ deposition precedes disease onset by more than two decades (Bateman et al, 2012). Thus, APP-Tg mice and *App* KI mice, which accumulate mainly full-length Aβ1-42, but little Aβ3pE-42, fail to exhibit major subsequent pathologies (tau pathology and neurodegeneration) even after humanization of the entire murine *Mapt* gene (Hashimoto et al, 2019; Saito et al, 2019). Because it is not feasible to spatiotemporally analyze the deposition of Aβ3pE-42 in the human brain, we can only make an assumption based on such mouse model data. Yang et al showed by MALDI-TOF mass spectrometry the presence of various Aβ peptides in different AD brains, whereas *App* KI mice uniformly accumulated Aβ1-42 (Yang et al, 2022), suggesting that Aβ1-42 undergoes truncation over decades after pathological deposition in the human brain. They also found "Type I and II filaments" in AD brain and only of "Type II" filaments in the *App* KI mouse brain. Because both Aβ1-42 and Aβ3pE-42 are abundant in AD brains (Fig S1) and because Aβ1-42 is the predominant variant in *App* KI mouse brain (Figs 6B and S9), Types I and II are likely composed of Aβ3pE-42 and Aβ1-42, respectively. Although this assumption needs to be experimentally validated, it may explain why Aβ in *App* KI or APP-Tg mice can be extracted by GuHCl (Kawarabayashi et al, 2001; Saito et al, 2014), whereas that in AD brains requires formic acid for Aβ extraction (Harigaya et al, 1995; Saido et al, 1995; Kawarabayashi et al, 2001). The difference in the quantity of Aβ3pE-42 accumulated in the human brain and in the mouse model remains an important issue to be solved.

Glutamate cyclization of Aβ3E-42 is a definite final process that results in Aβ3pE-42 production (Fig S10), and it is possible to regulate the production of Aβ3pE-42 by pharmacological or dietary means (Hennekens et al, 2015; Coimbra et al, 2019; Vijayan & Zhang, 2019). Alternatively, the ubiquitously present pyroglutamyl peptidase activity (Cummins & O'Connor, 1998), which was also down-regulated by *Mme* deficiency (Fig 7E), would convert Aβ3pE-42 to Aβ4F-42. In any event, it is advantageous that both Aβ3E-42 and Aβ4F-42 are even more labile to in vivo catabolism and less pathologically prone to deposition in the brain than the physiological Aβ1-42 form (Fig 1 and Table 1). Also, selective activation of NEP in the cortex and hippocampus is another therapeutic strategy (Saito et al, 2005; Watamura et al, 2022, 2024; Hori et al, 2023).

Lastly, our observations suggest that there may exist an "N-end rule" for Aβ catabolism. The original "N-end rule" for intracellular protein catabolism is closely associated with a specific class of ubiquitin ligases that recognize the amino-terminal structure of substrate proteins (Dougan et al, 2012; Kwon & Ciechanover, 2017; Dissmeyer et al, 2018). In this study, we demonstrated that the amino-terminal structure of Aβ influences its rate of metabolism. Although it is not yet clear how generalizable this "N-end rule" is for extracellular protein catabolism, there may exist a common mechanism that could account for the catabolism of relatively large and hydrophobic peptides represented by Aβ.

## Materials and Methods

### Radiolabeled peptides

Synthesis, purification, structural integrity, and subsequent in vivo analysis of radiolabeled Aβ peptides were performed as described for $^3$H/$^{14}$C-Aβ1-42 (Iwata et al, 2000). The Aβx-42 variants including Aβ1-42 were synthesized simultaneously from the carboxyl-terminus. When the sequential coupling of Fmoc amino acids proceeded to the point where Aβ17L-42 was produced, a fraction of the peptide–resin complex was isolated and subjected to deprotection and cleavage from the resin. The remaining portion of the peptide–resin was used to further synthesize Aβ4F-42, Aβ3E-42, Aβ3pE-42, Aβ2A-42, and Aβ1(D-Asp)-42 in a similar manner. Acetylated Aβ1-42 (Aβ1[Ac-Asp]-42) was produced by reacting Aβ1-42–containing resin with acetic anhydride after removal of the Fmoc group by piperazine. After extensive deprotection and cleavage with TFA, ethanedithiol, anisole, and dimethyl sulfide from the resin, followed by filtration, the peptides were washed three times by cold ether precipitation and subjected to size-exclusion chromatography (Shodex OHpak SB-802.5 HQ; Showa Denko) equilibrated with dimethylformamide. After removal of the solvent in the presence of an azeotropic solvent under vacuum, the purified peptides were dissolved in dimethyl sulfoxide and stored in aliquots at –80°C until use. The structural integrity of the peptides was confirmed by reversed-phase HPLC analysis (where the peptides were eluted as single peaks), peptide mapping with lysyl endopeptidase, direct chemical sequencing, and mass spectrometric analysis.

### Antibodies

Anti-Aβ antibodies specific to amino-terminal structures or carboxyl-terminal structures of Aβ variants were prepared as described previously (Saido et al, 1996) and were used for Western blot (WB) analysis and immunohistochemical staining (IHC) at the concentration of 1 μg/ml. N1D and N3pE, which were used widely throughout the present study, are rabbit polyclonal antibodies specific to the amino-terminals of Aβ1(D)-x and Aβ3(pE)-x, respectively. Other antibodies used in the present study are as follows: anti-pan Aβ antibody (mouse monoclonal 4G8, 1 μg/ml for WB, 2.5 μg/ml for IHC; Millipore), anti-ApoE (goat polyclonal, sc-6384, 0.4 μg/ml for WB; Santa Cruz Biotechnology), and rabbit anti-α1-antichymotrypsin (AP15343PU-N, 0.2 μg/ml for WB; Acris Antibodies).

### Human materials

Frozen brain tissues from the temporal cortex were obtained from the Manchester Brain Bank and frozen/stored at –80°C. Experiments using the human materials (Western blotting) have been approved by the Ethical Committee of RIKEN.

### Animals

All animal experiments were performed in compliance with the institutional guidelines of RIKEN and National Institutes for

Quantum Science and Technology. Male, 8–10-wk-old, Sprague Dawley rats used for in vivo analysis of metabolic rates of radio-labeled Aβx-42 peptides were obtained from Japan SLC. *Mme*-KO mice (Lu et al, 1995) and APP-Tg mice (APP23) (Sturchler-Pierrat et al, 1997) were kindly provided by Dr. Craig Gerard (Harvard Medical School) and Dr. Matthias Staufenbiel (Novartis Institutes for Biomedical Research), respectively. APP-Tg mice with a hetero-zygous *Mme*-deficient background (*Mme*$^{+/-}$APP-Tg) were produced by crossbreeding APP23 mice, which overexpress human-type APP-carrying double mutations (KM670/671NL), and homozygous *Mme*-deficient mice (*Mme*$^{-/-}$); *Mme*$^{+/+}$APP-Tg and *Mme*$^{-/-}$APP-Tg lit-termates were produced by breeding female *Mme*$^{+/-}$APP-Tg and male *Mme*$^{+/-}$APP-Tg mice, as described previously (Iwata et al, 2013). All mice were on the same genetic background (C57BL/6J). Both male and female mice were analyzed at the ages indicated in the figure legends. In the same way, *Mme*$^{+/+}$*App* KI (*App*$^{NL-F}$) mice and *Mme*$^{-/-}$*App*$^{NL-F}$ littermates were produced using *App*$^{NL-F/NL-F}$ KI mice (Saito et al, 2014). They were housed in plastic cages. One to three rats per cage (one to five mice per cage) received food (CE2; Clea Japan, Inc.) and water ad lib, and were maintained on a 12/12-h light–dark cycle (lights on at 09:00, off at 21:00).

## Development of *App*$^{NL-(ΔDA)-F}$ and *App*$^{NL-(ΔDA)-Q-F}$ KI mice

These mouse strains were developed according to the production method of *App*$^{NL-F}$ KI mice (Saito et al, 2014). First, we modified the targeting vector used to generate the *App*$^{NL-F}$ KI mice. In *App*$^{NL-(ΔDA)-F}$ KI mouse strain, the genes encoding two amino acids (aspartic acid and alanine at positions 1 and 2 in the Aβ sequence) were deleted from the targeting vector. In *App*$^{NL-(ΔDA)-Q-F}$ KI mouse strain, the two amino acids were similarly deleted and one point mutation from glutamic acid to glutamine at position 3 in the Aβ sequence was added to the targeting vector. Using these targeting vectors, we performed the same manipulations as in the devel-opment of the *App*$^{NL-F}$ KI mice. The first-generation mice obtained by gene targeting were crossed with EIIa-Cre transgenic mice (JAX: 003724; The Jackson Laboratory) to remove the neo resistance gene by the Cre-loxP system, and then backcrossed at least five gen-erations to establish a C57BL/6J genetic background as previously described (Saito et al, 2014).

## Immunohistochemistry and its quantitative assessment

Mouse brains were fixed by transcardial perfusion with phosphate-buffered 4% paraformaldehyde and then embedded in paraf-fin. Four-micrometer-thick sections were mounted onto aminoprop yltriethoxysilane-coated glass slides. The brain sections were im-munostained using anti-Aβ antibodies and visualized by the avidin–biotin–peroxidase complex procedure (Vectastain ABC kits; Vector Laboratories) using 3,3-diaminobenzidine as a chromogen. Sections were observed with a light microscope, Provis AX80 (Olympus Optical), using a 4X objective and a 1.5X digital zoom, and digital images were captured with a DP70 digital microscope camera (Olympus Optical). The density of immunoreactive Aβ deposits in the hippocampal formation and neocortex was measured using image analysis software, MetaMorph, ver. 6.1 (Universal Imaging Corporation, Downingtown, USA), after the raw images had been

inverted using image-editing software (Adobe Photoshop CS5; Adobe Systems, Inc.) by an investigator blinded as to sample identity. To reduce the variance among tissue sections, we used the average of data from three sections per mouse as an individual value.

Alternatively, amyloid pathology was detected using biotinylated secondary antibody and tyramide signal amplification (Akoya Biosciences). Before mounting, the sections were treated, when required, with Hoechst 33342 diluted in PBS. Data images were obtained using NanoZoomer Digital Pathology C9600 (Hamamatsu Photonics) and EVOS M5000 Imaging System (Thermo Fisher Sci-entific). Immunoreactive signals were quantified by Definiens Tis-sue Studio (Definiens).

## Lysyl endopeptidase (Lys-C) digestion and mass spectrometry

Monoisotopic mass values of Aβ variants in the detergent-insoluble/formic acid–soluble fractions were measured by a Microflex II MALDI-TOF mass spectrometer (Bruker Daltonics) in reflector mode, using standard operating parameters after the digestion of endoproteinase Lys-C, as previously described (Moore et al, 2012). Samples were dissolved with 70% (vol/vol) formic acid and subjected to sonication for 3 min. The formic acid was then evaporated, and the residue was dissolved in 20 μl of unbuffered 10 mM $NH_4HCO_3$ solution, subjected to sonication, and digested with 15 ng of endoproteinase Lys-C (Fujifilm Wako Chemicals) for 15 h at 37°C. The resulting peptide fragments were adsorbed on a MonoTip mini C18 pipette tip (GL Sciences, Inc.), desalted with 5% (vol/vol) methanol containing 0.1% (vol/vol) TFA, and eluted with 5 μl of 60% (vol/vol) acetonitrile containing 0.1% (vol/vol) TFA. The eluate was mixed with an equal volume of a 2,5-dihydroxybenzoic acid solution (10 mg/ml in 50% [vol/vol] acetonitrile; Tokyo Chemical Industry Co.), and 1 μl thereof was spotted on a stainless-steel Bruker MALDI 96 target plate (Bruker Daltonics). A baseline correction and anti-aliasing of mass spectra were carried out using FlexAnalysis 2.2 (Bruker Daltonics). External calibration of MALDI-TOF mass was carried out using singly charged monoisotopic peaks of peptide standards.

## Western blotting

Soluble and detergent-insoluble/formic acid–extractable fractions were prepared from mouse brains, and protein concentrations were determined using a BCA protein assay kit (Pierce). An equivalent amount of protein from each fraction was separated by 5–20% gradient SDS–polyacrylamide gel electrophoresis and transferred electrophoretically to 0.22-μm nitrocellulose mem-branes (Protoran, Whatman GmbH). The membranes were boiled in PBS for 3 min to gain high sensitivity. The blot was probed with anti-Aβ antibodies, followed by HRP-conjugated anti-mouse or anti-rabbit IgG (GE HealthCare Japan). An immunoreactive band on the membrane was visualized with an enhanced chemiluminescence kit (GE HealthCare Japan), and the band intensity was determined with a densitometer, LAS4000 (GE HealthCare Japan), using Science Lab 97 Image Gauge software (ver. 3.0.1; GE HealthCare Japan). Immunoreactive protein content in each sample was calculated based on a standard curve constructed with synthesized Aβ

(Bachem AG; Peptide Institute, Osaka, Japan). Each set of experiments was repeated at least three times. Formic acid–extractable fractions from neocortices of early-onset AD and late-onset AD patients were prepared according to the method described previously (Saido et al, 1995).

## Aβ ELISA

Mouse cerebral cortices were homogenized in buffer A (50 mM Tris–HCl, pH 7.6, 150 mM NaCl, and protease inhibitor cocktail) using Multi-Beads Shocker (Saito et al, 2014). The homogenized samples were directed to centrifugation at 200,000$g$ for 20 min at 4°C, and the supernatant was collected as a Tris-buffered saline (TS)–soluble fraction. The pellet was loosened with buffer A, centrifuged at 200,000$g$ for 5 min at 4°C, and then dissolved in 6 M guanidine HCl (GuHCl) buffer. After incubation at room temperature for 1 h, the sample was sonicated at 25°C for 1 min. Subsequently, the sample was centrifuged at 200,000$g$ for 20 min at 25°C and the supernatant collected as a GuHCl fraction. The amounts of Aβx-40, Aβx-42, Aβ1-40, Aβ1-42, Aβ3pE-40, and Aβ3pE-42 in each fraction were determined by sandwich ELISA kits: Fujifilm Wako #292-62301, Fujifilm Wako #298-62401, and Immuno-Biological Laboratories [IBL] #27713, #27711, #27418, and #27716, respectively.

## Autoradiographic and immunohistochemical labeling and PET imaging

Radiosynthesis of [$^{11}$C]PiB was conducted as described elsewhere (Maeda et al, 2007). Six-$\mu$m-thick paraformaldehyde-fixed brain sections from 24-mo-old $Mme^{+/+}$APP-Tg, $Mme^{+/-}$APP-Tg, and $Mme^{-/-}$APP-Tg mice were incubated in 50 mM Tris–HCl buffer (pH 7.4) containing 5% ethanol and [$^{11}$C]PiB (37 MBq/l, ~200 pM) at room temperature for 60 min. After the reaction, the samples were rinsed with ice-cold Tris–HCl buffer twice for 2 min and dipped into ice-cold water for 10 s. The slices were subsequently dried under warm blowing air and contacted to an imaging plate (Fuji Photo Film) for 1 h. Imaging plate data were scanned by a BAS5000 system (Fuji Photo Film). The intensities of radioactive signals in the neocortex and hippocampus were measured by Multi Gauge software (Fuji Photo Film). Specific binding of [$^{11}$C]PiB to amyloid in these regions was calculated as the difference in signal intensity versus the cerebellum, which lacks marked plaque lesions.

After attenuation of radioactivity, brain sections used for the autographic assay were reacted with rabbit polyclonal antibody against Aβ3pE (dilution, 1:50,000) (Saido et al, 1995, 1996), and immunolabeling was developed with a tyramide signal amplification-Direct kit (Akoya Biosciences).

Longitudinal PET scans were conducted for $Mme^{+/+}$APP-Tg, $Mme^{+/-}$APP-Tg, and $Mme^{-/-}$APP-Tg mice at 9, 13, and 20 mo of age, as stated elsewhere (Tai et al, 2005), with a microPET Focus 220 animal scanner (Siemens Medical Solutions USA) designed for rodents and small monkeys, which provides 95 transaxial slices 0.815 mm (center-to-center) apart, a 19.0-cm transaxial field of view, and a 7.6-cm axial field of view (Tai et al, 2005). Before the scans, the mice were anesthetized with 1.5% (vol/vol) isoflurane. After transmission scans for attenuation correction

using a 68Ge-68Ga point source, emission scans were performed for 60 min in a 3D list mode with an energy window of 350–750 keV, immediately after the intravenous injection of [$^{11}$C]PiB (32.0 ± 7.1 MBq). All list-mode data were sorted into 3D sinograms, which were then Fourier-rebinned into 2D sinograms (frames, 10 × 1, 8 × 5, and 1 × 10 min). Summation images from 30 to 60 min after [$^{11}$C]PiB injection were reconstructed with maximum a posteriori reconstruction, and dynamic images were reconstructed with filtered back-projection using a 0.5-mm Hanning filter. Volumes of interest were placed on multiple brain areas using PMOD image analysis software (PMOD Group) with reference to an MRI template generated in a previous study (Maeda et al, 2007). The radioligand binding in the hippocampus and neocortex was quantified by calculating the binding potential (a combined measure of the density of target molecules and affinity of the tracer) based on the specific binding compared with the non-displaceable uptake (BP$_{ND}$) using the simplified reference tissue model (Lammertsma et al, 1996) and cerebellar time–radioactivity curve as a reference.

## Immunopurification of peptidases

Coupling of Dynabeads protein G (Life Technologies Japan) with anti-mouse APA/BP1/6C3 antigen antibody, anti-mouse APN/CD13 antibody, or anti-mouse DPP4/CD26 antibody was carried out using 20 mM dimethyl pimelimidate dihydrochloride in 0.2 M triethanolamine (pH 8.2) according to the manufacturer's instructions. A Triton X-100–solubilized membrane fraction from neocortices of mice was precleared with Dynabeads protein G, and then incubated with each antibody coupled with Dynabeads protein G overnight at 4°C. The supernatant was removed from the peptidase–antibody–protein beads complex, and after several passages of washing, the peptidase was eluted from the complex with 25 mM citrate buffer (pH 3) into a tube containing 0.5N NaOH solution to neutralize pH. Immunopurified peptidases presented a single band on silver-stained SDS–PAGE; confirmation that they were desired peptidases was obtained by characterization based on substrate specificity, inhibitor sensitivity, and molecular mass on SDS–PAGE (data not shown).

## Measurement of peptidase activities

Two methods consisting of an Aβ digestion assay using HPLC and a fluorometric assay using artificial substrates were used.

To assay Aβ-degrading activity by APA or APN, a reaction mixture consisting of immunopurified mouse APA (120 pg) or mouse APN (400 pg), Aβ$_{1-40}$ (2 $\mu$g; Bachem AG), 1 mM CaCl$_2$, cOmplete, EDTA-free (Roche Diagnostics), 0.1 mM ZLLLal (Peptide Institute, Osaka, Japan), 10 $\mu$M pepstatin A (Peptide Institute), and 50 mM Tris–HCl (pH 7.5) in a total volume 50 $\mu$l was incubated for 5 h at 37°C, and terminated by the addition of 450 $\mu$l of 0.1% TFA. For the assay of Aβ-degrading activity by DPP4, 5 mM EDTA instead of 1 mM CaCl$_2$ and cOmplete, EDTA-free and immunopurified mouse DPP4 (58 pg) instead of the above peptidases were added to the above reaction mixture. Amounts of remaining intact Aβ and its proteolytic fragments were analyzed by HPLC (HP 1100 series; Agilent Technologies) connected to a UV detector. A reversed-phase polymer column (Capcell Pak

C18 UG120 Column; Shiseido) heated to 50°C and equilibrated with 0.1% TFA was used to separate peptides. Bound peptides were eluted by a linear gradient of 0–70% acetonitrile/0.1% TFA from 10 to 40 min at a flow rate of 0.2 ml/min. The APA- or APN-dependent Aβ-degrading activity was determined based on the decrease in the rate of digestion caused by 0.1 mM amastatin, a specific inhibitor of APA and APN. Protein concentrations were determined using a BCA protein assay kit (Pierce). For DPP4, 2 mM PMSF, a specific inhibitor of DPP4, was employed instead of amastatin. The activity was represented by the amount of degradation of μmol of a substrate/mg protein of enzyme/min.

APA-, APN-, and DPP4-dependent peptidase activities in Triton X-100–solubilized membrane fractions from neocortices of mice were fluorometrically assayed in the same assay mixtures as above except that 0.2 mM Glu-MCA, 0.1 mM Ala-MCA, and Gly-Pro-MCA (Bachem), respectively, were used as a substrate. Activities were determined from the fluorescence intensity at Ex. 390 nm and Em. 460 nm, based on the decrease in the rate of digestion caused by 0.1 mM amastatin or 2 mM PMSF, as mentioned above.

### RNA isolation and quantitative real-time PCR (qRT-PCR)

Total RNA was extracted from hippocampi of 18-mo-old male $Mme^{+/+}$ APP-Tg and $Mme^{-/-}$ APP-Tg mice using High Pure RNA Tissue Kit (Roche Diagnostics GmbH). After first-strand cDNA was synthesized using PrimeScript reverse transcriptase–PCR Kit (Takara Bio Inc.), qRT-PCR was conducted using Premix Ex Taq (Takara Bio Inc.) and a SmartCycler II system (Takara Bio Inc.) with the following two-step amplification profile: 40 cycles of 5 s at 95°C and 20 s at 60°C. Relative quantification of $Qpct$ (gene symbol for QC) and $Qpctl$ (gene symbol for QCL) RNA expression was carried out by normalization to the expression of glyceraldehyde 3-phosphate dehydrogenase (GAPDH). Nucleotide sequences of the probes and forward/reverse primers were Mm_Qpct-Probe (ACCAACCAACTGGATGGCATGGATCTGT) and Mm_Qpct-F (CCTCACCCTCCTGGATCAAGA)/Mm_Qpct-R (GGGAATGTTGGATTTGCTGCTC) for mouse $Qpct$, Mm_Qpctl-Probe (AGGCTACGTTGCAGTCCCTGTCGGC) and Mm_Qpctl-F (GGCAATCTCCAAGTGAGAAAGTTC)/Mm_Qpctl-R (GGGTCCAGTTCAACATGCCAG) for mouse $Qpctl$, and Mm_GAPDH-Probe (TGGTGGACCTCATGGCCTACATGGCC) and Mm_GAPDH-F (CAAGCTCATTTCCTGGTATGACAA)/Mm_GAPDH-R (TTGGGATAGGGCCTCTCTTGC) for mouse $Gapdh$.

### Statistics

All data are shown as the mean ± SEM. For comparisons of the means among three or more groups, statistical analysis was performed by one-way, two-way, or repeated-measures two-way analysis of variance (ANOVA) followed by a *post hoc* test; the Student–Newman–Keuls or the Tukey test was applied if the data passed the Shapiro–Wilk normality and the Brown–Forsythe equal variance tests. If the normality test or the equal variance test was not passed at ANOVA, the Kruskal–Wallis one-way ANOVA on the ranks was performed. These analyses were performed using SigmaPlot software ver.14.0 (Systat Software Inc) or GraphPad Prism 8 software (GraphPad software). $P$-values below 0.05 ($P < 0.05$) were considered to be significant throughout this study.

## Data Availability

All raw data presented in the article and supplementary information are available upon request from the corresponding authors.

## Supplementary Information

## Acknowledgements

We thank Taisuke Tomita, University of Tokyo, for valuable discussion. We also thank Yukiko Nagai-Watanabe for secretarial work. This work was supported by research grants from RIKEN, Special Coordination Funds for Promoting Science and Technology of STA, CREST, Ministry of Health and Welfare, Ministry of Education, Science and Technology, Chugai Pharmaceutical Co., Mitsubishi Chemical Co., Takeda Chemical Industries, and AMED under Grant Number JP20dm0207001 (Brain Mapping by Integrated Neurotechnologies for Disease Studies [Brain/MINDS]) (to TC Saido) and JSPS KAKENHI Grant Numbers JP18K07402 (to H Sasaguri), JP17025046 (to N Iwata), JP18023037 (to N Iwata), and JP20023031 (to N Iwata).

### Author Contributions

N Iwata: conceptualization, resources, data curation, software, formal analysis, supervision, funding acquisition, validation, investigation, visualization, methodology, project administration, and writing—original draft, review, and editing.

S Tsubuki: resources, data curation, software, formal analysis, investigation, visualization, methodology, and writing—original draft.

M Sekiguchi: resources, data curation, investigation, and methodology.

K Watanabe-Iwata: resources, data curation, formal analysis, investigation, and methodology.

Y Matsuba: resources, data curation, investigation, and methodology.

N Kamano: resources, data curation, investigation, and methodology.

R Fujioka: resources, data curation, investigation, and methodology.

R Takamura: data curation, formal analysis, investigation, and methodology.

N Watamura: data curation, formal analysis, validation, investigation, visualization, and methodology.

N Kakiya: data curation, validation, investigation, and methodology.

N Mihira: resources, data curation, investigation, and methodology.

T Morito: resources, data curation, formal analysis, investigation, and methodology.

K Shirotani: resources, data curation, formal analysis, investigation, and methodology.

DMA Mann: resources, data curation, investigation, and methodology.

AC Robinson: resources, data curation, investigation, and methodology.

S Hashimoto: resources, data curation, investigation, and methodology.

H Sasaguri: data curation, formal analysis, and methodology.

T Saito: conceptualization, resources, data curation, formal analysis, investigation, methodology, and writing—original draft.

M Higuchi: conceptualization, resources, data curation, formal analysis, supervision, investigation, visualization, methodology, and writing—original draft.

TC Saido: conceptualization, funding acquisition, validation, and writing—original draft, review, and editing.

## Conflict of Interest Statement

The authors declare that they have no conflict of interest.

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
