## [Reviewer comments · Life Science Alliance]

Metabolic resistance of A β 3pE-42, target epitope of donanemab

Takaomi Saido, Nobuhisa Iwata, Satoshi Tsubuki, Misaki Sekiguchi, Kaori Watanabe-Iwata, Yukio Matsuba, Naoko Kamano, Ryo Fujioka, Risa Takamura, Naoto Watamura, Naomasa Kakiya, Naomi Mihira, Takahiro Morito, Keiro Shirotnani, David Mann, Andrew Robinson, Shoko Hashimoto, Hiroki Sasaguri, Takashi Saito, and Makoto Higuchi

DOI: <https://doi.org/10.26508/lsa.202402650>

Corresponding author(s): Takaomi Saido, RIKEN Center for Brain Science

Review Timeline:

Submission Date:	2024-02-08
Editorial Decision:	2024-04-02
Revision Received:	2024-07-17
Editorial Decision:	2024-08-05
Revision Received:	2024-09-03
Editorial Decision:	2024-09-04
Revision Received:	2024-09-06
Accepted:	2024-09-06

Transaction Report:

April 2, 2024

Re: Life Science Alliance manuscript #LSA-2024-02650-T

Prof. Takaomi C. Saido
RIKEN Center for Brain Science
Laboratory for Proteolytic Neuroscience
Wako, Saitama 351-0198
Japan

Dear Dr. Saido,

Thank you for submitting your manuscript entitled "Metabolic resistance of A β 3pE-42, epitope of the anti-Alzheimer therapeutic antibody, donanemab.". The manuscript has been evaluated by expert reviewers, whose reports are appended below. Unfortunately, after an assessment of the reviewer feedback, our editorial decision is against publication in Life Science Alliance.

Although your manuscript is intriguing, I feel that the points raised by the reviewers are more substantial than can be addressed in a typical revision period. If you wish to expedite publication of the current data, it may be best to pursue publication at another journal.

Given the interest in the topic, I would be open to re-submission to Life Science Alliance of a significantly revised and extended manuscript that fully addresses the reviewers' concerns and is subject to further peer review. If you would like to resubmit this work to Life Science Alliance, you may submit an appeal directly through our manuscript submission system. Please note that priority and novelty would be reassessed at re-submission.

Regardless of how you choose to proceed, we hope that the comments below will prove constructive as your work progresses.

Thank you for thinking of Life Science Alliance as an appropriate place to publish your work.

Sincerely,

Reviewer #1 (Comments to the Authors (Required)):

Iwata and colleagues provide a manuscript on the metabolic resistance of A β 3pE-42, representing an abundant A β peptide variant that has been identified in the brains of patients suffering from Alzheimer's disease (AD). The authors claim that A β 3pE-42 is more stable than other A β variants and that the lack of a major A β -degrading protease induces A β 3pE-42 accumulation in a relative selective manner. The authors used several mouse lines to investigate this issue.

Several concerns should be addressed:

- p. 4, top paragraph: "supplementary Figure 2" should read Supplementary Figure 3 if I understood correctly.
- p. 4: "A β 3pE-42, is the most abundant in both early- and late-onset cases". This statement falls short and is not correct as e.g. A β 4-42 has been shown to represent a highly abundant species (Masters et al., 1985) but has not been assessed in the current panel.
- p.4: "This specific form accounts for >50% of total A β X-42 as quantitated against varying amounts of synthetic A β peptides, in agreement with Kuo et al..." As said before, other reports state that e.g. 4-x represent more than 60% of A β peptides found in plaque cores (Masters et al., PNAS 1985). This should be mentioned as well.
- p. 5, metabolic analysis of peptide stability in rat brains: Is this somewhere described in the material and methods part? Did the authors inject fresh or aggregated peptides?
This is crucial information as it might have direct implications on the interpretation of the data. It has been shown in Rostagno et al., Transl Neurodegen 2022 that monomeric A β species are readily cleared from the brain, while e.g. aggregated 4-40 and 4-42 are much more resistant. Given that pGlu3-42 is a fast aggregating species, could the outcome of the experiment be influenced by the aggregation state? The authors have to provide data on the aggregation of their injected peptide species and

experimental details need to be added to the manuscript.

- p. 5: "Again, these truncated A β peptides without pyroglutamate are consistently seen as minor components deposited in the senile human brain (Supplementary Figure 1)". This is not shown in Fig. S1 as 2-42 and 4-42 are not analysed.

p. 5: "The only exception is A β 17L-42 (p3 fragment): " Why are no retention time plots shown for Abeta1D-Asp-42 and Abeta 17L-42?"

p. 7: "NEP deficiency increased not only the histochemical quantity but also the biochemical quantity of A β 3pE-42 in the APP-Tg brain (Figure 4)". It will probably also increase the biochemical quantity of other species such as 4-16 which have not been analysed as the spectra are cropped at 1700 m/z. Full spectra should be shown and all variants should be annotated.

p. 7: Why do the authors use different assays to compare Abeta1-42 and N3pE-42 levels in APP23 and APP knock-in lines (Western-blot in APP23 and sandwich ELISA in APP knock-in)? To make clear-cut assumptions, all animals should be analysed with the sandwich ELISA in my opinion, which would also be a more direct comparison.

What is the biochemical quantity of N3pE-42 compared to 1-42 in the APP23 lines? This cannot be assessed with the western-blot data showing only relative values.

- p. 7, last paragraph: Why do the authors not observe any differences in N3pE-levels between mouse lines with an initial E or Q residue? Q should be a much better substrate for pGlu-conversion.

p. 8: "The relative amount of A β 3pE-42 per total A β in the App

knock-in mouse model is as small as less than 1% (Supplementary Figure 4)". To be correct it is only 0.1%.

p. 9: "...whereas the App knock-in mice uniformly accumulated A β 1-42 (Yang et al., 2022), suggesting that A β 1-42 undergoes truncation over decades after pathological deposition in the human brain." It should be discussed here, that the APP knock-in mice carry the Swedish mutation and are therefore an artificial model strongly favoring the generation of Asp1-Abeta species. This is not representative of the human situation and should not be used as a model to explain generation of truncated species in general.

Reviewer #2 (Comments to the Authors (Required)):

Iwata et al. analyzed the metabolic stability of synthetic A β species, and the deposition of several A β species derived from the amyloid precursor protein in transgenic mouse models, and human brain samples. Consistent with previous studies, there is a considerable difference in the relative amount of pyroglutamate modified A β (3pE-A β 42) in human and mouse brains. The data indicate a role of the N-terminus in the determination of metabolic stability and deposition of A β variants. The study is of potential interest, but several data sets appear preliminary, and descriptive in nature. Overall, the organization and writing of the manuscript could be improved. Some conclusions are not well supported by the provided data.

Specific comments:

1. The introduction and results section should be clearly indicated. The major aim of the study should be clearly stated. In the text, authors shift between description of previous findings and present data.
2. Injection of radiolabeled synthetic A β into the hippocampal region of rats (Fig.1) indicate a higher metabolic stability of 3pE-A β 42 as compared to unmodified and several other A β species with different N-termini. This is interesting. However, the speculation about an N-end rule for extracellular peptides/proteins appears quite far-fetched. To support a more general role of N-terminal amino acids in determination of peptide stability more comprehensive experiments would be required, e.g. using peptides of similar length with different N-terminal amino acids.
3. Could the differential stability of injected A β species be affected by their aggregation characteristics. Inclusion of A β 40 species in the injection experiments could address this issue. The uncleaved peptides (Fig.1) are indicated by arrowheads. The elution profile (pattern of peaks) between 0 and 20 min differs between the respective peptides. This could indicate different degradation products deriving from the different peptide variants. These should be analyzed by mass spectrometry.
4. Are there different number of data points in figure 1 g-h or do data points overlap? E.g. 1H, only 1 data point at 60 min.? The exact number of experiment/replicates should be mentioned in the figure legend.
5. Authors write that „It is also notable that the catabolic intermediate identified during the A β 1-42 degradation (A β 10-37)(Iwata et al., 2000) is missing in the proteolysis of A β 2-42, A β 3E-42 and A β 3pE-42, but not of A β 1(Ac-Asp)-42 (Figure 1). Is A β 10-37 detected here?"
6. It would be important to control aggregation state of A β before injection and after incubation. E.g. by separation on native gels
7. Authors write that „Supplementary Figure 1 shows that, among the structural variants known to be present in AD brains, the A β 42 bearing amino-terminal pyroglutamate, A β 3pE-42, is the most abundant in both early- and late-onset cases". Is this conclusion solely based on the blot provided in this figure? It is not appropriate to make such conclusions from WB data with different antibodies that might have (very) different affinities and 'cross-reactivities'. Signals for A β 40/42/43 are very weak if detected at all. Authors should explain.
8. To support a role of NEP in the degradation of the different A β species, injection experiments could be done in wt and NEP KO mice.
9. Number of mice differ between 4-10 for different sex and genotype groups. Authors should explain.
10. Sentence „It is notable the heterozygous Mme deficiency, which corresponds to approximately 50% NEP activity reduction (Iwata et al, 2001), resulted in a relative increase in the N3pE/N1D ratio because the NEP expression invariably declines with aging (Hellstrom-Lindahl et al, 7 2008; Iwata et al, 2002; Russo et al, 2005; Wang et al, 2003)." reads confusing.
11. Fig. 2: why only female mice analyzed for plaque size (fig.2c-d)
12. Authors should explain terms 'solid-state' and 'solid-liquid interface'. Do they mean plaques and oligomers?

13. In fig.3 several peaks are shown in the Mm^e-/-APP-Tg sample (2B). But only two were assigned as pE1-16 and E1-16. Could the neighboring peaks also be assigned?
14. Fig. 4A: some dotted lines are missing in the upper panel? Why only female mice were analyzed here?
15. It is apparent from this figure that that signal for 0.1 ng A β 3pE-42 detected with antibody N3pE is much stronger than that for 4 ng A β 1-40/42 detected with the other antibodies, indicating considerable differences in the sensitivity of the different antibodies under these conditions. This could interfere with the quantification of the different A β species in brain extracts and IHC, and requires consideration, also with respect to the statement „A β 3(pE)-42 is invariably the predominant species all the cases" in the legend to supplement Figure1S.
16. What is meant with „Two-way ANOVA revealed that significant interactions between nerilysin-deficiency and ages..."
17. Authors write that „Incidentally, NEP deficiency also accelerated the deposition of both A 1-42 and A 3pE-42 in the AppNL-F line of App knock-in mice, but there was no significant difference in the ratio of A 3pE 42/A 1-42." Is this shown in the manuscript ?
18. It is mentioned that „In any case, histochemical analysis of APP-Tg (Figure 2) and App knock-in lines (Supplementary Figure 4 & 5) consistently indicated that deposition of A 3pE-42 takes place much later than that of A 1-42." However, authors compare the deposition of A β 1-x and A 3pE-X (supplementary figure 4A). ELISA detection only revealed co-deposition of A x-40/42 and A 3pE-40/42 (supplement fig. 4B). Sex of mice is not indicated in this figure. The calibration of the ELISA should be described in more detail.
19. Authors should check and correct the labeling and description of panels in Fig. 5 and Fig.6.
Additional points:
20. Authors should explain the meaning of terms „endo-specific antibodies", „strictly physiological form"
21. It is mentioned that „whereas A 1-40 and A 1-42 are the major species secreted by cells(Scheuner et al, 1996; Suzuki et al, 1994)". Here, the cell types should be mentioned. It should also be mentioned that the generation of different truncated A species (and of p3) could vary between cell types.

RESPONSES TO REVIEWERS' CRITICISMS

We have highlighted the revised sentences in green according to your comments and highlighted revised sentences in yellow to improve the manuscript.

Reviewer #1 (Comments to the Authors (Required)):

Iwata and colleagues provide a manuscript on the metabolic resistance of A β pE3-42, representing an abundant A β peptide variant that has been identified in the brains of patients suffering from Alzheimer's disease (AD). The authors claim that A β pE3-42 is more stable than other A β variants and that the lack of a major A β -degrading protease induces A β pE3-42 accumulation in a relative selective manner. The authors used several mouse lines to investigate this issue.

Several concerns should be addressed:

1. p. 4, top paragraph: "supplementary Figure 2" should read Supplementary Figure 3 if I understood correctly.

Thank you for your careful review and suggestion. We have corrected the citation of figure.

2. p. 4: "A β 3pE-42, is the most abundant in both early- and late-onset cases". This statement falls short and is not correct as e.g. Abeta4-42 has been shown to represent a highly abundant species (Masters et al., 1985) but has not been assessed in the current panel.

3. p.4: "This specific form accounts for >50% of total A β X-42 as quantitated against varying amounts of synthetic A β peptides, in agreement with Kuo et al..." As said before, other reports state that e.g. 4-x represent more than 60% of Abeta peptides found in plaque cores (Masters et al., PNAS 1985). This should be mentioned as well.

5. p. 5: "Again, these truncated A β peptides without pyroglutamate are consistently seen as minor components deposited in the senile human brain (Supplementary Figure 1)". This is not shown in Fig. S1 as 2-42 and 4-42 are not analysed.

7. p. 7: "NEP deficiency increased not only the histochemical quantity but also the biochemical quantity of A β 3pE-42 in the APP-Tg brain (Figure 4)". It will probably also increase the biochemical quantity of other species such as 4-16 which have not been analysed as the spectra are cropped at 1700 m/z. Full spectra should be shown and all variants should be annotated.

In comments 2,3,5 and 7, the reviewer queries why we did not assess A β 4-42, which was previously

detected as a highly abundant variant (Masters, et al., EMBO J. 1985 Nov;4(11):2757-63; Proc Natl Acad Sci USA. 1985 Jun;82(12):4245-9). In these studies, the amino acid sequence of A β was determined by using a protein sequencer. In this method, the α -amino group of N-terminal amino acid is coupled with phenyl isothiocyanate, followed by Edman degeneration, and then probes for HPLC analysis. In this case, a modified amino acid, in which the amino group at the α -position is blocked, is not coupled with phenyl isothiocyanate, hindering sequence analysis. We assume that they removed such a modified amino acid to determine the sequence technically, which is the reason why they did not detect A β 3pE-x. In the present study, we did not detect mass spectra including A β 4-16 (1639.75 m/z) smaller than 1700 m/z in the formic acid-extractable fraction of the brain, as shown in the following mass spectrometric profile. We have added the profiles (m/z 1480-1940, m/z 4000-4600) as Supplementary Figure 4S and the theoretical mass and detected mass in Supplementary Table 3. Signals from wild-type mice, which were regarded as background, were subtracted from those of *Mme*^{+/+}APP-Tg and *Mme*^{-/-}APP-Tg mice. The mass signals without annotation were not related to A β .

Mass spectrometric profiles between m/z 1480 and 1940.

Mass spectrometric profiles between m/z 4000 and 4600.

To respond to your concern regarding our statement "*A β 3pE-42*, is the most abundant in both early- and late-onset cases" as the basis of our supporting data (Supplementary Figure 1) and previous observations, we quantified signal intensities of immunoreactive bands on the blots using standard curves composed of variable amounts of synthetic A β and have added the quantitative data to Supplementary Figure 1, where the amount of A β 3pE-x comprised 41.5% of total A β , but that of A β 1-x only 4.8%. These data support the premise of our study and the development of the drug donanemab.

On this basis, we corrected the sentence "*This specific form accounts for >50% of total A β x-42*" to "*This specific form accounts for the majority of total A β x-42*". Kawarabayashi *et al.* (*J Neurosci* 21(2):372–381, 2001) also reported a result that is consistent with ours. We have attached these data here.

Amount of A β x42- comprised 92% of total A β when focused on the C-terminus of A β .

4. p. 5, metabolic analysis of peptide stability in rat brains: Is this somewhere described in the material and methods part? Did the authors inject fresh or aggregated peptides? This is crucial information as it might have direct implications on the interpretation of the data. It has been shown in Rostagno et al., Transl Neurodegen 2022 that monomeric Abeta species are readily cleared from the brain, while e.g. aggregated 4-40 and 4-42 are much more resistant. Given that pGlu3-42 is a fast aggregating species, could the outcome of the experiment be influenced by the aggregation state? The authors have to provide data on the aggregation of their injected peptide species and experimental details need to be added to the manuscript.

We have described the method for using radiolabeled peptides in the first paragraph of the **Supplemental Materials and Methods** section.

After extensive deprotection and cleavage with trifluoroacetic acid, we collected a fraction, including synthetic A β and subjected it to size-exclusion chromatography equilibrated with dimethylformamide. The purified peptides were then analyzed for uniformity and eluted as single peaks corresponding to “the uncleaved peptides at 0 min,” as shown in Figure 1. The synthetic A β s were dissolved in dimethylsulfoxide, divided into several tubes to avoid freeze-thaw, and kept at -80°C. Thus, we confirmed that a single peak (uncleaved peptide at 0 min) showed the monomeric form in each case.

6. p. 5: "The only exception is A β 17L-42 (p3 fragment): " Why are no retention time plots shown for Abeta1D-Asp-42 and Abeta 17L-42?"

In the original manuscript, the retention time plots are included in Figure 1, which has now been moved to Supplementary Figure 3.

Please refer to the plots.

8. p. 7: Why do the authors use different assays to compare Abeta1-42 and N3pE-42 levels in APP23 and APP knock-in lines (Western-blot in APP23 and sandwich ELISA in APP knock-in)? To make clear-cut assumptions, all animals should be analysed with the sandwich ELISA in my opinion, which would also a more direct comparison.

We carried out additional experiments according to your comment. New data showing A β 1-42 and A β 3pE-42 levels in APP23, quantitated using sandwich ELISAs, are now included in Figure 4D. Please refer to the following data comparing A β levels between APP23 and *App*-KI.

	A β 3pE-x:	A β 1-x:	ratio of A β 3pE-x/A β 1-x
Fig.4D: APP23, 24M	0.947 nmol/g tissue	148 nmol/g tissue	0.0064
Fig.8S: App -KI, 24M	1.204 pmol/g tissue	2.66 nmol/g tissue	0.00046

9. *What is the biochemical quantity of N3pE-42 compared to 1-42 in the APP23 lines? This cannot be assessed with the western-blot data showing only relative values.*

In the original manuscript, we described the quantitative values of immunoreactive A β variants using calibration curves composed of synthetic A β in the legend of Figure 4B and 4C as follows, “The amounts of immunoreactive A β variants in 24-month-old *Mme*^{+/+}APP-Tg mouse brains to N1D, N3pE, 4G8, C40 and C42 are 15.80, 0.028, 10.12, 8.18 and 1.70 ng/ μ g protein in (B), and 241.3, 1.150, 895.7, 461.6 and 91.7 ng/ μ g protein in (C), respectively.” Please refer to this description in the text.

In the AD brains as shown in Supplementary Figure 1, the concentration of N1D (A β 1-x) is 0.0427 \pm 0.0205 ng/ μ g protein, that of N3pE (A β 3pE-x) is 0.3691 \pm 0.0518 ng/ μ g protein. The ratio of A β 3pE-x/A β 1-x is 8.64. The relative amount of A β 3pE-x to A β 1-x in AD brains was greater than that in APP-Tg mouse brains.

10. *p. 7, last paragraph: Why do the authors not observe any differences in N3pE-levels between mouse lines with an initial E or Q residue? Q should be a much better substrate for pGlu-conversion.*

As mentioned there, it was unexpected that neither the *App*^{NL-(ADA)-F} nor the *App*^{NL-(ADA)-Q-F} lines exhibited any visible A β pathology, even after aging (i.e., 18 months or more) (**Figure 6C**). However, because in the brains of 2-month-old *App*^{NL-(ADA)-Q-F} mice, where a trace amount of A β 3pE-x was detected, Q appears to be a much better substrate for pGlu-conversion in the mouse brain. As A β 2-x and 3-x are metabolically unstable and A β 3pE-x has been detected in the mouse brain, we hypothesized that N-truncation and cyclization of Glu may give rise to a cooperative process where dipeptidyl aminopeptidase and glutaminyl-peptide cyclotransferase are involved.

11. *p. 8: "The relative amount of A β 3pE-42 per total A β in the *App* knock-in mouse model is as small as less than 1% (Supplementary Figure 4)". To be correct it is only 0.1%.*

Thank you for your careful review and suggestion. We have corrected this statement accordingly.

12. *p. 9: "...whereas the *App* knock-in mice uniformly accumulated A β 1-42 (Yang et al., 2022), suggesting that A β 1-42 undergoes truncation over decades after pathological deposition in the*

human brain." It should be discussed here, that the APP knock-in mice carry the swedish mutation and are therefore an artificial model strongly favoring the generation of Asp1-Abeta species. This is not representative of the human situation and should not be used as a model to explain generation of truncated species in general.

Our research purpose is to clarify how A β 1-x deviated from “on-pathway”, which is a physiological pathway mediated by neprilysin, is catabolized, and accumulated in the brain in the basis of observations of A β species accumulated in AD brains and in several lines of AD model mice. Neprilysin does not affect A β generation, including any effects of the Swedish mutation, as shown in many previous studies published by us and other laboratories.

To the best of our knowledge, the Swedish mutation has been used for almost all lines of AD model mice, including APP23, to increase A β generation and accumulation in the brains. The *App*-KI mouse is a much better AD model as it excludes artifacts that may be caused by transgenes, as observed in APP-Tg mice. The *in vivo* experimental paradigm using radiolabeled A β variants would complement analyses using AD model mice.

Reviewer #2 (Comments to the Authors (Required)):

Iwata et al. analyzed the metabolic stability of synthetic A β species, and the deposition of several A β species derived from the amyloid precursor protein in transgenic mouse models, and human brain samples. Consistent with previous studies, there is a considerable difference in the relative amount of pyroglutamate modified A β (3pE-A β 42) in human and mouse brains. The data indicate a role of the N-terminus in the determination of metabolic stability and deposition of A β variants. The study is of potential interest, but several data sets appear preliminary, and descriptive in nature. Overall, the organization and writing of the manuscript could be improved. Some conclusions are not well supported by the provided data.

Specific comments:

1. The introduction and results section should be clearly indicated. The major aim of the study should be clearly stated. In the text, authors shift between description of previous findings and present data.

We apologize for the irregular format of the descriptions in this manuscript. We have rearranged the structure of these sections.

2. Injection of radiolabeled synthetic A β into the hippocampal region of rats (Fig.1) indicate a higher metabolic stability of 3pE-A β 42 as compared to unmodified and several other A β species with different N-termini. This is interesting. However, the speculation about an N-end rule for extracellular peptides/proteins appears quite far-fetched. To support a more general role of N-terminal amino acids in determination of peptide stability more comprehensive experiments would be required, e.g. using peptides of similar length with different N-terminal amino acids.

We have toned down the statement regarding the N-end rule for extracellular peptides/proteins because we do not have enough data to hypothesize such a role. However, the N-terminal structure of A β affects metabolic rates, as shown in this study. We revised this statement to emphasize this fact.

3. Could the differential stability of injected A β species be affected by their aggregation characteristics. Inclusion of A β 40 species in the injection experiments could address this issue. The uncleaved peptides (Fig.1) are indicated by arrowheads. The elution profile (pattern of peaks) between 0 and 20 min differs between the respective peptides. This could indicate different degradation products deriving from the different peptide variants. These should be analyzed by mass spectrometry.

First, A β 1-42 is more relevant to AD pathogenesis than is A β 1-40.

Next, chemical synthesis of peptides starts to couple Fmoc-amino acids from the C-terminus of

the peptide to the N-terminus; therefore, we could prepare multiple radiolabeled A β variants, including nine radiolabeled Fmoc amino acids. Because \$8,000 is required to purchase one radiolabeled Fmoc amino acid, it is difficult to synthesize radiolabeled A β 1-40 and its variants right now. Peaks between 0 and 20 min in the elution profile of each A β variant include free amino acids and very short peptides that are degraded nonspecifically by various peptidases; thus, we consider that it is not meaningful to further analyze them.

4. Are there different number of data points in figure 1 g-h or do data points overlap? e.g. 1H, only 1 data point at 60 min.? The exact number of experiment/replicates should be mentioned in the figure legend.

We have added a statement regarding the number of data points to the legend of Figure 3S. Each analysis of the *in vivo* catabolic rate of the A β x-42 variant was carried out in independent experiments with more than five time points.

We have also revised the last sentence in the legend of Figure 1 as follows: Digitized data were used to calculate the *in vivo* half-lives of the peptides, as shown in Supplementary Figure 3 and Table 1.

5. Authors write that “It is also notable that the catabolic intermediate identified during the A β 1-42 degradation (A β 10-37)(Iwata et al., 2000)” is missing in the proteolysis of A β 2-42, A β 3E-42 and A β 3pE-42, but not of A β 1(Ac-Asp)-42 (Figure 1). Is A β 10-37 detected here?

After carefully re-checking the data in the panels B-F of Figure 1, we found that tiny peaks which may be the catabolic intermediate A β 10-37, were observed at a retention time of 37 min. We have added an illustration of arrowheads indicating the intermediate in the panels in Figure 1, and a sentence “A black arrowhead shows A β 10-37 in the panel 1A (A β 1-42), and green arrowhead show possible intermediates in other panels” to the legend. Thank you for your comments.

p.6, line 13 **in the original manuscript**: We deleted the sentence starting from “It is also notable that the catabolic intermediate ….”

p.5, lines 21-23 **in the revised manuscript**: We added a sentence as follows “In any case, NEP appears to contribute to catabolism, because a catabolic intermediate similar to A β 10-37 detected during A β 1-42 degradation (Iwata et al., 2000) was observed (Figure 1, green arrowheads).”

6. It would be important to control aggregation state of A β before injection and after incubation. e.g. by separation on native gels.

We collected a fraction, including synthetic A β , after extensive deprotection and cleavage with trifluoroacetic acid and subjected this to size-exclusion chromatography equilibrated with dimethylformamide. The purified peptides were then analyzed for uniformity and eluted as single peaks corresponding to “the uncleaved peptides at 0 min,” as shown in Figure 1. The synthetic A β s were dissolved in dimethylsulfoxide, divided into several tubes to avoid freeze-thaw and kept at -176 °C. We confirmed that single peak (uncleaved peptide at 0 min) showed the monomeric form in each case.

7. Authors write that „Supplementary Figure 1 shows that, among the structural variants known to be present in AD brains, the A β 42 bearing amino-terminal pyroglutamate, A β 3pE-42, is the most abundant in both early- and late-onset cases“. Is this conclusion solely based on the blot provided in this figure? It is not appropriate to make such conclusions from WB data with different antibodies that might have (very) different affinities and ,cross-reactivities'. Signals for A β 40/42/43 are very weak if detected at all. Authors should explain.

We quantified signal intensities of immunoreactive bands on the blots using standard curves composed of variable amounts of synthetic A β and have added the quantitative data to Supplementary Figure 1, where the amount of A β 3pE-x comprised 41.5% of total A β , but that of A β 1-x only 4.8%. The data support the premise of our study and the development of donanemab. Therefore, we corrected the sentence "This specific form accounts for >50% of total A β x-42" to "This specific form accounts for the majority of total A β x-42".

8. To support a role of NEP in the degradation of the different A β species, injection experiments could be done in wt and NEP KO mice.

In vivo experiments analyzing amyloid deposition in the brains of AD model mice can complement the experiments commented by this reviewer.

9. Number of mice differ between 4-10 for different sex and genotype groups. Authors should explain.

There was a limitation in the number of aged mice available for analysis. For example, we must mate more than 20 pairs of heterozygote mice to obtain one genotype at the age of 24 months, keeping in mind the associated housing costs and space for more than three years. Another reason is that the timing of generation and crossbreeding of the mice was different among some lines.

10. Sentence “It is notable the heterozygous Mme deficiency, which corresponds to approximately

50% NEP activity reduction (Iwata et al, 2001), resulted in a relative increase in the N3pE/N1D ratio because the NEP expression invariably declines with aging (Hellstrom-Lindahl et al, 7 2008; Iwata et al, 2002; Russo et al, 2005; Wang et al, 2003)." reads confusing.

We revised the sentence as follows,
“Given that NEP expression invariably declines with aging (Hellstrom-Lindahl et al., 7 2008; Iwata et al., 2002; Russo et al., 2005; Wang et al., 2003), it is notable that even heterozygous *Mme* deficiency, presenting an approximately 50% reduction in NEP activity (Iwata et al, 2001), resulted in a relative increase in the ratio of N3pE to N1D.”

11. Fig. 2: why only female mice analyzed for plaque size (fig.2c-d)

In general, female mice accumulated A β much faster than male mice, whereas *Mme* deficiency did not affect sex differences, as analyzed using 2-way ANOVA (Figs. 2B and 2C). If male mice were used to analyze plaque size, similar results could be obtained. Another reason is that the sample size of female mice was greater than that of male mice. We are usually able to obtain more aged female mice than aged male mice because male mice often die as a result of fighting during aging.

12. Authors should explain terms, 'solid-state' and 'solid-liquid interface'. Do they mean plaques and oligomers?

Exactly. We revised the phrases as follows, “a solid-state such as the core of amyloid plaques and in a liquid-solid transition state such as plaque formation from soluble A β monomers and oligomers.”

13. In fig.3 several peaks are shown in the Mme-/-APP-Tg sample (2B). But only two were assigned as pE1-16 and E1-16. Could the neighboring peaks also be assigned?

In general, mass spectrometry signals of peptides are detected as clusters of signals because stable isotopes are present at a constant rate. When a relatively weak signal in the mass spectrum graph is magnified, we can see them. In addition, during sample preparation, such as extraction by formic acid, a peptide often undergoes oxidation [M+16], formylation [M+28], addition of sodium [M+22], etc. In Fig. 3B. the difference in the mass of the neighboring peaks is less than two, and they are isotropic variations. In general, the annotation of such variations is not included in a profile.

14. Fig. 4A: some dotted lines are missing in the upper panel? Why only female mice were analyzed here?

Thank you for this suggestion. It appears that the dotted lines are not easy to see. We have revised this illustration. Female mice were analyzed for the reasons described above.

15. It is apparent from this figure that that signal for 0.1 ng A β 3pE-42 detected with antibody N3pE is much stronger than that for 4 ng A β 1-40/42 detected with the other antibodies, indicating considerable differences in the sensitivity of the different antibodies under these conditions. This could interfere with the quantification of the different A β species in brain extracts and IHC, and requires consideration, also with respect to the statement". A β 3(pE)-42 is invariantly the predominant species all the cases" in the legend to supplement Figure 1S.

To solve this issue, we performed multidirectional analysis using different experimental paradigms, such as quantitative WB, ELISA, and image analysis for immunohistochemistry. Standard curves composed of varying amounts of synthetic peptides were used for biochemical quantification of A β .

16. What is meant with "Two-way ANOVA revealed that significant interactions between nerilysin-deficiency and ages..."

We suppose that *Mme* deficiency-dependent accelerated A β accumulation affected any factor relevant to the generation of A β 3(pE)-42, e.g. up-regulation of APN, DPP4 and QPCTL as shown in Figure 7.

17. Authors write that "Incidentally, NEP deficiency also accelerated the deposition of both A β 1-42 and A β 3pE-42 in the AppNL-F line of App knock-in mice, but there was no significant difference in the ratio of A β 3pE 42/A β 1-42." Is this shown in the manuscript?

We have added the results in Figure 6.

18. It is mentioned that "In any case, histochemical analysis of APP-Tg (Figure 2) and App knock-in lines (Supplementary Figure 4 & 5) consistently indicated that deposition of A β 3pE-42 takes place much later than that of A β 1-42." However, authors compare the deposition of A β 1-x and A β 3pE-X (supplementary figure 4A). ELISA detection only revealed co-deposition of A β x-40/42 and A β 3pE-40/42 (supplement fig. 4B). Sex of mice is not indicated in this figure. The calibration of the ELISA should be described in more detail.

Please note the differences in the units of the Y-axes between A β x-40/42 and A β 3pE-40/42. We have added new data to compare the ratio of A β 3pE-x to A β x-40/42 in the brains of 15- and 24-month-old *App*^{NL-F} KI mice (Supplementary Figure 8C). The ratio increased at a later age.

We included information on the sex of the mice: “Numbers of analyzed animals were as follows: 1 female and 2 male WT, 1 female and 2 male *App*^{NL} KI, and 2 female and 2 male *App*^{NL-F} KI mice.”

19. Authors should check and correct the labeling and description of panels in Fig. 5 and Fig.6.

We apologize for this confusion. We have revised the labels and descriptions.

Additional points:

20. Authors should explain the meaning of terms “endo-specific antibodies”, “strictly physiological form”.

We changed from “endo-specific antibodies” to “**end**-specific antibodies”, and “the strictly physiological form” to “physiologically secreted form”.

21. It is mentioned that “whereas A β 1-40 and A β 1-42 are the major species secreted by cells (Scheuner et al, 1996; Suzuki et al, 1994)”. Here, the cell types should be mentioned. It should also be mentioned that the generation of different truncated A β species (and of p3) could vary between cell types.

It is a well-known fact on the basis of accumulated data from Alzheimer’s disease studies that A β 1-40 and A β 1-42 are the major species secreted by cells. It is not necessary to specify the cell types. We have cited a review article regarding A β generation.

August 5, 2024

Re: Life Science Alliance manuscript #LSA-2024-02650-TR-A

Prof. Takaomi C. Saido
RIKEN Center for Brain Science
Laboratory for Proteolytic Neuroscience
2-1 Hirosawa
Wako, Saitama 351-0198
Japan

Dear Dr. Saido,

Thank you for submitting your revised manuscript entitled "Metabolic resistance of A β 3pE-42, target epitope of the anti-Alzheimer therapeutic antibody, donanemab" to Life Science Alliance. The manuscript has been seen by the original reviewers whose comments are appended below, and some important issues remain.

Our general policy is that papers are considered through only one revision cycle; however, we are open to one additional short round of revision. Please note that I will expect to make a final decision without additional reviewer input upon re-submission.

Please submit the final revision within one month, along with a letter that includes a point by point response to the remaining reviewer comments.

To upload the revised version of your manuscript, please log in to your account: <https://lsa.msubmit.net/cgi-bin/main.plex>
You will be guided to complete the submission of your revised manuscript and to fill in all necessary information.

B. MANUSCRIPT ORGANIZATION AND FORMATTING:

Sincerely,

Reviewer #1 (Comments to the Authors (Required)):

My comments were sufficiently addressed.

Reviewer #2 (Comments to the Authors (Required)):

In the revised manuscript, authors added new data and explanations. However, the present data mainly provide only indirect or no support of the authors' conclusions. Direct demonstration of decreased cleavage of A β 3pE-42 by NEP is missing (e.g. by in vitro assays using purified NEP and the different A β species). Several statements are not well supported by the data, and larger parts of the manuscript read confusing. The manuscript requires much more accurate phrasing, careful interpretation of data, and more rigorous experimental planning.

Specific comments:

- Authors should carefully check the logic within individual and connected sentences. (e.g. "However, our observations of in vivo A β 1-42 catabolism demonstrated that specific endoproteolysis mediated by neprilysin (NEP), but not aminopeptidase action, is the major rate-limiting step with A β 10-37 as a catabolic intermediate (Iwata et al, 2000). These observations are better explained by assuming that different A β forms have different life spans, for which the structural determinants may reside in the amino-terminal residues of the peptide." or "The present study aims to examine whether the amino-terminal structure of A β x-42 influences its catabolism in vivo based on its structural alteration in the AD brain." , and several more.

- The new sentence "It is likely produced in a solid-state such as the core of amyloid plaques and in a liquid-solid transition state such as plaque formation from soluble A β monomers and oligomers." in the abstract reads confusing. A similar phrasing is used in the results section: "This observation implies that the A β 3E-x -to- A β N3pE conversion may take place in a solid-state such as the core of amyloid plaques and in a liquid-solid transition state such as plaque formation from soluble A β monomers and oligomers".

- No references are given for the first two statements/sentences in the introduction.

- Authors write that "The selective deposition of this physiologically rare A β species in human brain can be attributed to its presumed metabolic stability because pyroglutamy peptide is resistant to major aminopeptidases except for pyroglutamy aminopeptidase (Mori et al., 1992)". However, the cited reference does not support metabolic stability of A β 3pE-42.

- It is stated that "We found that a deficiency of NEP, the major A β -degrading enzyme, activated the compensatory pathway, where aminopeptidases, dipeptidyl peptidase and glutaminy-peptide cyclotransferase are involved,..". Is this shown in the manuscript?

- What is the meaning of "isogenic A β x-42 variants" ?

- What is meant with "direct or indirect interactions between A β and NEP, which has a catalytic site cavity with a diameter of only approximately 20 angstroms (Moss et al, 2018, 2020)."

- This reviewer noted in the first review that "The elution profile (pattern of peaks) between 0 and 20 min differs between the respective peptides. This could indicate different degradation products deriving from the different peptide variants. These should be analyzed by mass spectrometry." Authors respond that "Peaks between 0 and 20 min in the elution profile of each A β variant include free amino acids and very short peptides that are degraded nonspecifically by various peptidases; thus, we consider that it is not meaningful to further analyze them." What is meant with "...are degraded nonspecifically by various peptidases"? I think it would be important to characterize the additional degradation products to support the specific involvement of NEP in the injection experiments. In the same vein, how could authors identify A β (10-37) in the eluate as a cleavage intermediate?(fig.1A). What are "possible intermediate" labeled by green arrowheads?

- The size distribution of plaques is surprising (fig. 2c). The highest number of plaques is sized between 50 and 200 μ m. Then there is an additional "peak" of plaques sized between 1.500 and 2.500 μ m. Is there an explanation for this? Why did authors not show a graph for N3pE positive plaques larger than 10.500 μ m (as done for N1D positive plaques)?

- Authors write that "N3pE immunoreactivity tended to be present in smaller/cored plaques (<300 μ m²), whereas N1D was present in larger plaques (9,500-20,000 μ m²)." As mentioned above the graphs in fig. 2c show highest number of N1D positive plaques sized between 50 and 200 μ m, and an additional major peak of plaques sized between 1.500 and 2.500 μ m. Overall, a similar distribution of plaque size is observed for N3pE immunoreactivity. The question remains why only female mice of one age were used for plaque size analysis (fig.2c), while analyses of plaque area (fig.2a,b,c) were done with male and female mice of different age groups. This is not rigorous.

- Are the indicated significance values in fig.2c for the complete distribution or for individual plaque sizes? No explanation is given for the rectangle in the left graph for N3pE (fig.2c). It does not seem to be appropriate to compare the +/+ cohort with combined cohorts of +/- and -/- groups (fig. 2c, N3pE, left graph; and fig. 2d) or the -/- cohort with combined +/+ and +/- cohorts (fig.2c, N1D, right graph).

- What is meant with "a significant main effect ..."? legend to fig. 2d.

- Is there a different meaning of "A β 3pE" and "A β N3pE" in the text?

- Authors state that "Notably, Mme-deficiency increased the amounts of A β N1D and A β N3pE in middle-aged mice,...". However, a significant difference in N1D is only indicated in the detergent-insoluble fraction at 12M of age. The sex of the mice is not indicated (fig. 4).

- Why colocalization analysis with PiB was only done for N3pE, not for N1D species? Again, this is not rigorous.

- Authors state that "In any case, both histochemical and biochemical analyses of APP-Tg (Figures 2, 4 & 5) and AppNL-F line

of App K1 mice (Figure 6B, Supplementary Figure 8C) consistently indicated that deposition of A β 3pE-42 takes place much later than that of A β 1-42." However, there is no support for this conclusion.

RESPONSES TO REVIEWERS' CRITICISMS

We have highlighted the revised sentences in yellow according to the reviewer's comments.

Reviewer #2 (Comments to the Authors (Required)):

*In the revised manuscript, authors added new data and explanations. However, the present data mainly provide only indirect or no support of the authors' conclusions. Direct demonstration of decreased cleavage of A β 3pE-42 by NEP is missing (e.g. by *in vitro* assays using purified NEP and the different Abeta species). Several statements are not well supported by the data, and larger parts of the manuscript read confusing. The manuscript requires much more accurate phrasing, careful interpretation of data, and more rigorous experimental planning.*

We analyzed differences in *in vivo* catabolic rates of A β variants with various N-terminal structures and then focused on the relationship between deficits in neprilysin involved in *in vivo* catabolism of A β 1-42 and generation of metabolically resistant variant A β 3pE-42 using multiple experimental paradigms, such as radiolabeled A β variants and several lines of AD model mice.

Our major finding is that a significant reduction in NEP activity upregulated a compensatory A β degradation pathway composed of aminopeptidases, dipeptidyl peptidase, and glutaminyl-peptide cyclotransferase, and then part of A β 1-42 was processed to A β 3pE-42, which accumulated in the brain.

We determined differences in *in vivo* catabolic rates of N-terminal variants of A β using multiply radiolabeled A β as demonstrated for physiologically generated A β 1-42, but did not go far enough to clarify their metabolic processes. Therefore, we have toned down or deleted some sentences regarding the statement "neprilysin appears to be involved in *in vivo* cleavage of N-terminal variants of A β ." in response to the reviewers' comments.

Specific**comments:**

*- Authors should carefully check the logic within individual and connected sentences. (e.g. "However, our observations of *in vivo* A β 1-42 catabolism demonstrated that specific endoproteolysis mediated by neprilysin (NEP), but not aminopeptidase action, is the major rate-limiting step with A β 10-37 as a catabolic intermediate (Iwata et al, 2000). These observations are better explained by assuming that different A β forms have different life spans, for which the structural determinants may reside in the amino-terminal residues of the peptide." or "The present study aims to examine whether the amino-terminal structure of A β x-42 influences its catabolism *in vivo* based on its structural alteration in the AD brain.", and several more.*

We thank the reviewer for his/her comment. We have re-checked the description throughout the text to improve our manuscript, revised phrases and sentences that are not easy to understand, and added

new sentences as highlighted in green.

- The new sentence "It is likely produced in a solid-state such as the core of amyloid plaques and in a liquid-solid transition state such as plaque formation from soluble A β monomers and oligomers." in the abstract reads confusing. A similar phrasing is used in the results section: "This observation implies that the A β 3E-x -to- A β N3pE conversion may take place in a solid-state such as the core of amyloid plaques and in a liquid-solid transition state such as plaque formation from soluble A β monomers and oligomers".

We have revised the sentence and added supplementary explanation.

p.11, 2nd paragraph: Interestingly, we detected A β 3pE-16 as well as trace quantities of A β 2A-16 and A β 3E-16 in *Mme*^{-/-}APP-Tg mouse brain (**Figure 3B**). Furthermore, *Mme*-deficiency increased not only the histochemical quantity of amyloid plaques but also the biochemical quantity of A β 3pE-42 in the soluble fraction of APP-Tg brains as analyzed by WB and ELISA (**Figure 4 and Supplementary Figure 6**). The presence of A β 3pE-42 in a soluble form even prior to plaque formation in the human brain (Russo et al., 1997) indicates the presence of a dynamic equilibrium between the liquid and solid phases. These observations imply that the N-terminal truncation of A β 2A-x and conversion from A β 3E-x to A β 3pE-x may take place both in a solid-state such as the core of amyloid plaques and in a liquid-solid transition state, such as formation of amyloid plaques from soluble A β monomers and oligomers. A β 3pE-42-positive signals corresponding to PiB-binding sites were detected preferentially in the core of amyloid plaques, whereas A β 1-42-positive signals were observed in larger amyloid plaques (**Figure 5A**). If A β 3pE-42 generation occurred only in the solid-state, it is difficult to account for this morphological difference.

- No references are given for the first two statements/sentences in the introduction.

We added appropriate references.

- Authors write that "The selective deposition of this physiologically rare A β species in human brain can be attributed to its presumed metabolic stability because pyroglutamyl peptide is resistant to major aminopeptidases except for pyroglutamyl aminopeptidase (Mori et al., 1992)". However, the cited reference does not support metabolic stability of A β 3pE-42.

We have cited the ref (Mori et al., 1992) for "pyroglutamyl peptide is resistant to major aminopeptidases except for pyroglutamyl aminopeptidase".

- It is stated that "We found that a deficiency of NEP, the major A β -degrading enzyme, activated the compensatory pathway, where aminopeptidases, dipeptidyl peptidase and glutaminyl-peptide cyclotransferase are involved,..". Is this shown in the manuscript?

Please refer to the data from Figure 7, which had been added to the previous revised version of the manuscript.

- *What is the meaning of "isogenic Aβx-42 variants" ?*

We meant that the Aβx-42 variants are derived from Aβ1-42. However, it may not be easy for readers to understand the meaning of the term; therefore, we have deleted the term.

- *What is meant with "direct or indirect interactions between Aβ and NEP, which has a catalytic site cavity with a diameter of only approximately 20 angstroms (Moss et al, 2018, 2020)."*

We deleted the sentence.

- *This reviewer noted in the first review that "The elution profile (pattern of peaks) between 0 and 20 min differs between the respective peptides. This could indicate different degradation products deriving from the different peptide variants. These should be analyzed by mass spectrometry." Authors respond that "Peaks between 0 and 20 min in the elution profile of each Aβ variant include free amino acids and very short peptides that are degraded nonspecifically by various peptidases; thus, we consider that it is not meaningful to further analyze them." What is meant with "...are degraded nonspecifically by various peptidases"? I think it would be important to characterize the additional degradation products to support the specific involvement of NEP in the injection experiments. In the same vein, how could authors identify Aβ(10-37) in the eluate as a cleavage intermediate?(fig.1A). What are "possible intermediate" labeled by green arrowheads?*

We determined the eluate as Aβ10-37 using mass spectrometry and characterized as a cleavage intermediate in our previous report (Iwata *et al.*, 2000), therefore in the present study we observed a peak of the eluate at the identical retention time in the analysis using HPLC in each case, so we regarded this as "possible intermediate."

- *The size distribution of plaques is surprising (fig. 2c). The highest number of plaques is sized between 50 and 200 μm. Then there is an additional "peak" of plaques sized between 1.500 and 2.500 μm. Is there an explanation for this? Why did authors not show a graph for N3pE positive plaques larger than 10.500 μm (as done for N1D positive plaques)?*

Unfortunately, we have no evidence that accounts for the pattern of size distribution of the amyloid plaques. Because population of N3pE positive plaques larger than 10,500 μm was very small, compared with N1D-positive plaques, we combined the number of plaques larger than 10,500 μm and showed as >10,500 μm. This figure has now been revised.

- *Authors write that "N3pE immunoreactivity tended to be present in smaller/cored plaques (<300*

μm^2), whereas N1D was present in larger plaques (9,500-20,000 μm^2)." As mentioned above the graphs in fig. 2c show highest number of N1D positive plaques sized between 50 and 200 μm , and an additional major peak of plaques sized between 1.500 and 2.500 μm . Overall, a similar distribution of plaque size is observed for N3pE immunoreactivity. The question remains why only female mice of one age were used for plaque size analysis (fig.2c), while analyses of plaque area (fig.2a,b,c) were done with male and female mice of different age groups. This is not rigorous. - Are the indicated significance values in fig.2c for the complete distribution or for individual plaque sizes? No explanation is given for the rectangle in the left graph for N3pE (fig.2c). It does not seem to be appropriate to compare the +/+ cohort with combined cohorts of +/- and -/- groups (fig. 2c, N3pE, left graph; and fig. 2d) or the -/- cohort with combined +/+ and +/- cohorts (fig.2c, N1D, right graph).

Mme^{+/+}APP-Tg, 10 *Mme*^{+/-}APP-Tg; 8 *Mme*^{-/-}APP-Tg mice, the sum of the number of sections we analyzed is 150) and 18-months-old female mice (7 *Mme*^{+/+}APP-Tg, 10 *Mme*^{+/-}APP-Tg, 8 *Mme*^{-/-}APP-Tg mice, the sum of the number of sections is 90) in addition to that from 24-months-old female mice (the sum of the number of sections is 172), and have included the statistical data in the figure legend. The significance values showing in Fig. 2c (N3pE, left graph and N1D, right graph) and Fig. 2d indicate that the distribution patterns were significantly different in the three or two groups when compared using all cohorts of +/+, +/-, and -/- groups, based on the statistical results from two-way ANOVA. Therefore, this method is appropriate. We did not use combined cohorts from any group for statistical analysis. To avoid misunderstandings, we revised the illustration.

- What is meant with "a significant main effect ..."? legend to fig. 2d.

This is a statistical term that indicates that the ratios of N3pE to N1D-positive areas are significantly different in *Mme* genotypes.

- Is there a different meaning of "A β 3pE" and "A β N3pE" in the text?

We have unified the term "A β 3pE" to "A β 3pE"-x or "A β 3pE-42(40)" because N3pE is the name of antibody.

- Authors state that "Notably, *Mme*-deficiency increased the amounts of A β N1D and A β N3pE in middle-aged mice,...". However, a significant difference in N1D is only indicated in the detergent-insoluble fraction at 12M of age. The sex of the mice is not indicated (fig. 4).

We thank the reviewer for his/her careful review of our manuscript. We revised the sentence as follows, Page 8, lines 4-8: "*Mme*-deficiency prominently increased the amounts of A β 3pE-x from middle-age, but a difference in the amount of A β 1-x between the two genotypes was not observed

with the progression of age. In contrast, the amount of A β 3pE-x in both genotypes kept increasing until 24-month-old age (**Figure 4B and 4C**), resulting in an increased ratio of A β 3pE-40/42 to A β 1-40/42 (**Figure 4D, rightmost graph**).” The mice used in the analysis were female. We added this information in the legend.

- Why colocalization analysis with PiB was only done for N3pE, not for N1D species? Again, this is not rigorous.

We assessed increased deposition of A β 3pE-42 by *Mme*-deficiency *in vivo* by microPET based on colocalization of A β 3pE-positive plaques with PiB-binding sites. This is because we previously reported that A β 3pE-42-positive plaques, but not A β 1-42-positive plaques, colocalized with PiB-binding sites (Maeda et al., 2007).

- Authors state that "In any case, both histochemical and biochemical analyses of APP-Tg (Figures 2, 4 & 5) and AppNL-F line of App KI mice (Figure 6B, Supplementary Figure 8C) consistently indicated that deposition of A β 3pE-42 takes place much later than that of A β 1-42." However, there is no support for this conclusion.

In each case, the deposition of A β 3pE-42 was observed later than that of A β 1-42 in an aging-dependent assessment. We find this reviewer’s comment confusing.

September 4, 2024

RE: Life Science Alliance Manuscript #LSA-2024-02650-TRR

Prof. Takaomi C. Saido
RIKEN Center for Brain Science
Laboratory for Proteolytic Neuroscience
2-1 Hirosawa
Wako, Saitama 351-0198
Japan

Dear Dr. Saido,

Thank you for submitting your revised manuscript entitled "Metabolic resistance of A β 3pE-42, target epitope of donanemab". We would be happy to publish your paper in Life Science Alliance pending final revisions necessary to meet our formatting guidelines.

- please be sure that the authorship listing and order is correct
- please add the Twitter handle of your host institute/organization as well as your own or/and one of the authors in our system
- please make sure the author order in your manuscript and our system match
- please be sure that all authors are mentioned in the Author's Contribution section in the manuscript file
- please add your main, supplementary figure, and table legends to the main manuscript text after the references section;
- please upload your Tables in editable .doc or Excel format
- include references from the supplementary file in the references in the main manuscript file
- please label supplementary figures as S1, S2, etc.
- please add callouts for Figures 1A-F; 3A; 4A-C; 7B; S5A-C; S8A and table S2 to your main manuscript text

FIGURE CHECKS:

- please add scale bars to Figures 5A, 5B and S7A

A. FINAL FILES:

B. MANUSCRIPT ORGANIZATION AND FORMATTING:

Sincerely,

September 6, 2024

RE: Life Science Alliance Manuscript #LSA-2024-02650-TRRR

Prof. Takaomi C. Saido
RIKEN Center for Brain Science
Laboratory for Proteolytic Neuroscience
2-1 Hirosawa
Wako, Saitama 351-0198
Japan

Dear Dr. Saido,

Thank you for submitting your Research Article entitled "Metabolic resistance of A β 3pE-42, target epitope of donanemab". It is a pleasure to let you know that your manuscript is now accepted for publication in Life Science Alliance. Congratulations on this interesting work.

DISTRIBUTION OF MATERIALS:

Again, congratulations on a very nice paper. I hope you found the review process to be constructive and are pleased with how the manuscript was handled editorially. We look forward to future exciting submissions from your lab.

Sincerely,
